# An Ising model on a 3D honeycomb zigzag-ladder lattice: A solution to the ground-state problem

Yuriy I. Dublenych[1,2][*] and Oleg A. Petrenko[3]

**1** Institute for Condensed Matter Physics, National Academy of Sciences of Ukraine,
1 Svientsitskii Str., 79011 Lviv, Ukraine
**2** Max Planck Institute for the Physics of Complex Systems,
38 Nöthnitzer Str., 01187 Dresden, Germany
**3** Department of Physics, University of Warwick, Coventry CV4 7AL, United Kingdom

[*] dubl@icmp.lviv.ua

## Abstract

A complex, seven-parameter ground-state problem for an Ising model on a 3D honeycomb zigzag-ladder lattice, containing two types of magnetic sites, is considered in the presence of an external field using the method of basic rays and basic sets of cluster configurations. It is shown that the geometrical frustration due to the presence of triangle elements leads to the emergence of a large variety of magnetic phases, the majority of which are highly degenerate. The obtained theoretical results are used to elucidate the sequence of phase transitions in the family of rare-earth oxides with a honeycomb zigzag-ladder lattice, $SrRE_2O_4$ and $BaRE_2O_4$. New phases predicted by our model and observed experimentally do not appear in previously considered simpler models for non-interacting zigzag-ladders.

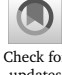
# 1   Introduction

Geometrically frustrated magnets, due to richness of their magnetic structures and behaviors, have been among the most intensively studied objects in the physics of magnetism and magnetic materials over the past several decades. Their theoretical description is a rather difficult task, especially in the case when quantum effects are essential. However, among frustrated magnets, there are many compounds with large-moment magnetic atoms. Often, these magnets can be well described with classical, either Heisenberg or Ising spin models, depending on the presence of strong crystal-field effects.

In this paper, we study geometrically frustrated magnets with magnetic atoms carrying large spins. These are 3D honeycomb zigzag-ladder magnets such as $SrRE_2O_4$ [1, 2] and $BaRE_2O_4$ [3–6], where RE is a rare earth atom. These families of compounds exhibit a very rich magnetic behavior, especially in an external magnetic field [7–15]. Rare earth magnetic atoms in these compounds occupy two crystallographically inequivalent positions with substantially different values of magnetic moments (which can be considered as classical) and, very often, almost orthogonal directions of easy-axis magnetization.

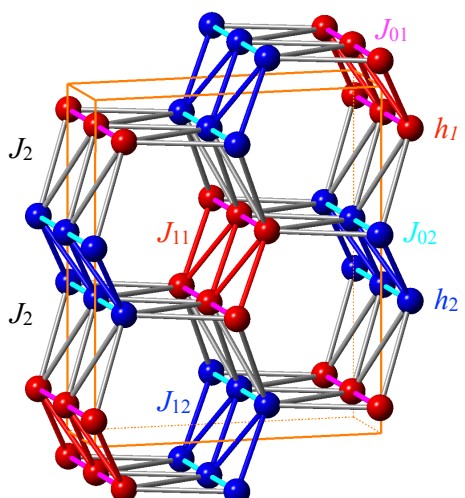

Figure 1: A honeycomb zigzag-ladder lattice comprised of two types of sites, red and blue. The coupling between the two neighboring spins along the ladder legs is $J_{01}$ for red sites and $J_{02}$ for blue sites. The spin coupling along ladder rungs is $J_{11}$ and $J_{12}$ for the red and blue sites, respectively, while the coupling between red and blue neighboring sites is $J_2$. The external field for the two sites is represented by $h_1$ and $h_2$. This magnetic lattice is found, for example, in the $SrRE_2O_4$ compounds [1] for which the orange box represents a crystallographic unit cell.

One of the most important experimental and theoretical challenges is to determine the magnetic structures of these magnets in an external magnetic field at low temperature. If an appropriate Hamiltonian is established, then one can try to solve the ground-state problem. This is difficult, even for rather simple classical Hamiltonians and, although several methods have been developed [16–27], in particular the method of geometric inequalities by J. Kanamori and M. Kaburagi [20, 21], no general algorithm exists to obtain the ground-state phase diagrams.

To appropriately describe the honeycomb zigzag-ladder magnets with strong magnetic anisotropy, we consider an Ising-like Hamiltonian with seven parameters (see Fig. 1). Since we deal with two types of spins, and the direction of the external field is arbitrary, two, rather than one, external field parameter needed to be introduced. Although we refer to spins throughout the text, one has to keep in mind that the orbital contribution to the magnetic moments is significant in almost all $SrRE_2O_4$ and $BaRE_2O_4$ compounds.

The ground-state problem for Ising-like Hamiltonians can be solved by using the method of basic rays and basic sets of cluster configurations [28–32]. This is the only method that gives the complete solution for a ground-state problem of Ising-like Hamiltonians, i.e, all the ground-state structures in every point of Hamiltonian parameter space. In the previous studies by one of the authors, this method was mostly used to rigorously prove the completeness of solutions [28, 31, 32]. Here, we use the method as a tool for finding a solution to the ground-state problem for an Ising model on a honeycomb zigzag-ladder lattice with two types of sites. Although we restrict our considerations to the smallest clusters (triangular plaquettes) the problem is rather complex because there are four types of such plaquettes with six configurations for each type. We have found 22 basic rays (edges of ground-state regions in the parameter space), but this is not a complete set – to determine all the basic rays, larger clusters should be considered. It might be possible to eventually establish a complete solution for this ground-state problem using an advanced, specially developed software package, however, even the incomplete solution found here sheds light on the sequence of magnetic transitions observed in the honeycomb zigzag-ladder magnets in the $SrRE_2O_4$ and $BaRE_2O_4$ families. Due to strong crystal-field effects, the magnetic moments in several members of these families demonstrate Ising-like behaviour with the easy-axes directions varying from one rare-earth site to another, which makes them a good fit to the model considered.

The paper is organized as follows. Subsection 2.1 of Section 2 gives a description of the model under consideration and the cluster method used. In Subsection 2.2, triangular plaquettes and their spin configurations are introduced, the Hamiltonian is presented as a sum of energies of all the plaquettes of the lattice. In Subsection 2.3, all the basic rays (vectors) which can be found using the triangular plaquettes are listed. Fully dimensional (that is, seven-dimensional) ground-state regions and corresponding ground-state structures found on the basis of these basic rays are described in Subsection 2.4. In Subsection 2.5, the disorder (degeneracy) of the fully dimensional phases is analyzed. In Subsection 2.6, "nontriangular" fully dimensional structures neighboring "triangular" ones are constructed and analyzed and, in Subsection 2.7, six examples of ground-state phase diagrams are presented. In Section 3, the relation between the experimental and the theoretical results is discussed and, in Section 4, conclusions are drawn.

## 2 "Triangular" ground-state structures

### 2.1 Model and method

The magnetic lattice of 3D honeycomb zigzag-ladder magnets is shown in Fig. 1. The structure is composed of three types of zigzag ladders: red, blue and gray (after the color of rungs in Fig. 1). We will refer to a part of the lattice with one red, one blue, and four gray ladders as a "hexagonal tube". There are two species of nonequivalent sites, these are depicted with two colors: blue and red. The value of spin at each site is equal to $-1$ or $+1$. The coupling between two neighboring spins along ladder legs (rungs) is $J_{01}$ for red sites and $J_{02}$ for blue ones ($J_{11}$ and $J_{12}$). The coupling between spins at neighboring sites of different colors (along ladder rungs) is $J_2$. There are also two external field parameters, $h_1$ and $h_2$ for the red and blue sites, respectively; these parameters depend on the components of an external field along the two easy magnetization axes and the values of the magnetic moments at red and blue sites. We therefore consider an Ising-type model with seven parameters and the Hamiltonian of the model reads

$$
\mathcal{H} = \sum_{\substack{\langle \text{magenta} \\ \text{bonds} \rangle}} J_{01}\sigma_i\sigma_j + \sum_{\substack{\langle \text{cyan} \\ \text{bonds} \rangle}} J_{02}\sigma_i\sigma_j + \sum_{\substack{\langle \text{red} \\ \text{bonds} \rangle}} J_{11}\sigma_i\sigma_j + \sum_{\substack{\langle \text{blue} \\ \text{bonds} \rangle}} J_{12}\sigma_i\sigma_j
$$
$$
+ \sum_{\substack{\langle \text{gray} \\ \text{bonds} \rangle}} J_2\sigma_i\sigma_j - \sum_{\substack{\langle \text{red} \\ \text{sites} \rangle}} h_1\sigma_i - \sum_{\substack{\langle \text{blue} \\ \text{sites} \rangle}} h_2\sigma_i . \tag{1}
$$

To find the ground states of such a model, we use a cluster method developed by one of the authors in previous papers, the so-called method of basic rays and basic sets of cluster configurations [28, 29, 31, 32]. Let us briefly elaborate on the main aspects of the method used.

The ground-state phase diagram for any Ising-type model is a set of convex polyhedral cones (polyhedral angles with the vertices at the origin of coordinates) in the parameter space. A polyhedral cone is the linear hull, that is, all linear combinations with nonnegative coefficients — the so-called conic hull — of a set of vectors. An $n$-dimensional polyhedral cone is bounded by $(n-1)$-, $(n-2)$-, ..., 2-, 1-faces. 1-faces are called edges. The polyhedral cone is fully determined by its edges (vectors along them). The most important are fully dimensional polyhedral cones (seven-dimensional for the model considered). These cones fill the parameter space without gaps and overlaps. We refer to a structure, which is a ground-state structure in a fully dimensional polyhedral cone, as fully dimensional and to the corresponding edges (vectors) as basic rays (vectors) [28–32]. A ground-state problem can be considered as resolved if all the edges (basic rays or basic vectors) of all the fully dimensional polyhedral cones are determined as well as all the ground states at these edges. The ground states in basic rays (the same along an entire ray) are constructed with the lowest energy configurations of a cluster (or clusters). We refer to the sets of these configurations as "basic sets of cluster configurations." Simple examples of basic rays and basic sets of cluster configurations are given in Refs. [31, 32] and in the appendix of Ref. [29]. It should be noted that the lower the dimension of a face, the more degenerate is the corresponding ground state. The most degenerate ground states correspond to 1-faces, i.e., edges.

### 2.2 Triangular plaquettes and their energies

Let us consider the simplest plaquettes of the lattice shown in Fig. 1 – triangular ones (Fig. 2). There are four types of triangular plaquettes, the total energy can be distributed between them in different ways, as every plaquette has vertices and sides shared with the neighboring plaquettes. The arbitrariness in energy distribution can be taken into account by introducing

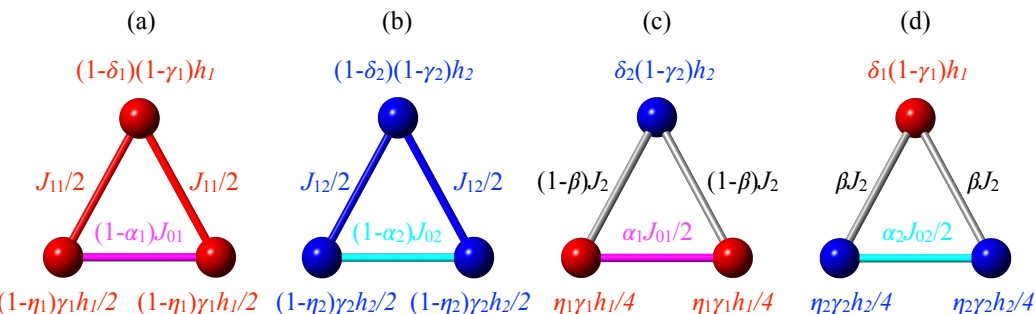

Figure 2: Four types of triangular plaquettes and their energies (see Fig. 1). The energy distribution between plaquettes of different types with shared sites or bonds is parameterized by a set of coefficients $\alpha_{1,2}$, $\beta$, $\gamma_{1,2}$, and, $\eta_{1,2}$, which take arbitrary values between zero and unity and which we refer to as "free" coefficients.

a set of coefficients $\alpha_1$, $\alpha_2$, $\beta$, $\gamma_1$, $\gamma_2$, $\eta_1$, and $\eta_2$ which can take arbitrary values between zero and one and which we refer to as "free" coefficients. The four types of the triangular plaquettes and energy distribution between them are shown in Fig. 2. The Hamiltonian (1) can be presented as a sum of energies for all the plaquettes,

$$
\begin{aligned}
\mathcal{H} = \sum_{i} &\left[ (1-\alpha_1)J_{01}\sigma_{i1}\sigma_{i2} + \frac{J_{11}}{2}(\sigma_{i2}\sigma_{i3}+\sigma_{i3}\sigma_{i1}) - \frac{1-\eta_1}{2}\gamma_1 h_1(\sigma_{i1}+\sigma_{i2}) \right.\\
&\left. -(1-\delta_1)(1-\gamma_1)h_1\sigma_{i3} \right]\\
+ \sum_{i} &\left[ (1-\alpha_2)J_{02}\sigma_{i1}\sigma_{i2} + \frac{J_{12}}{2}(\sigma_{i2}\sigma_{i3}+\sigma_{i3}\sigma_{i1}) - \frac{1-\eta_2}{2}\gamma_2 h_2(\sigma_{i1}+\sigma_{i2}) \right.\\
&\left. -(1-\delta_2)(1-\gamma_2)h_2\sigma_{i3} \right]\\
+ \sum_{i} &\left[ \frac{\alpha_1}{2}J_{01}\sigma_{i1}\sigma_{i2} + (1-\beta)J_2(\sigma_{i2}\sigma_{i3}+\sigma_{i3}\sigma_{i1}) - \eta_1\frac{\gamma_1}{4}h_1(\sigma_{i1}+\sigma_{i2}) - \delta_2\frac{1-\gamma_2}{2}h_2\sigma_{i3} \right]\\
+ \sum_{i} &\left[ \frac{\alpha_2}{2}J_{02}\sigma_{i1}\sigma_{i2} + (1-\beta)J_2(\sigma_{i2}\sigma_{i3}+\sigma_{i3}\sigma_{i1}) - \eta_2\frac{\gamma_2}{4}h_2(\sigma_{i1}+\sigma_{i2}) - \delta_1\frac{1-\gamma_1}{2}h_1\sigma_{i3} \right],
\end{aligned}
\tag{2}
$$

where the first, second, third, and fourth summations go over all the plaquettes shown in Fig. 2.

Let us show, for instance, that the energy of every red site is taken into account only once in the sum of energies of all the plaquettes on the lattice, that is, in the Hamiltonian (2). Every red site belongs to three plaquettes of type $a$ (to one in the upper position and to two in the lower positions), to four plaquettes of type $c$, and to two plaquettes of type $d$. Therefore the one-site energy is

$$
e = -\sigma h_1 \left( (1-\delta_1)(1-\gamma_1) + 2\frac{(1-\eta_1)\gamma_1}{2} + 4\frac{\eta_1\gamma_1}{4} + 2\frac{\delta_1(1-\gamma_1)}{2} \right) = -\sigma h_1,
\tag{3}
$$

where $\sigma$ is the value of spin at the red site. The Hamiltonian (2) does not depend on free coefficients despite the fact that the four sums for the individual plaquettes do depend on them.

There are six configurations of each plaquette, °°, °•, •°, ••, °•, and ••, where open and solid circles denote spins $\sigma = -1$ and $\sigma = +1$, respectively. The energies of these configurations for all the four types of plaquettes are given in Appendix A.

## 2.3 Basic rays and basic sets of triangular plaquettes configurations

Using the expressions for these energies (see Appendix A), one can find 22 basic rays. They are given in Table 1. In the first column of the table, the basic rays **r** [7-vectors $(J_{01}, J_{02}, J_{11}, J_{12}, J_2, h_1, h_2)$] are listed. Symbols $^\star, \tilde{}$, and $^-$ denote the following transformations: sublattices swap (red sublattice becomes blue and vice versa), spin flip on the blue sublattice, and spin flip on both sublattices. In the second column, the ground-state configurations of the four types of plaquettes for the corresponding basic ray are given. The symbol $\parallel$ separates configurations for four different types of triangular plaquettes. The symbol °° denotes the set of all the six configurations. Basic vector (ray) $\mathbf{r}_1^\star$, for instance, and the corresponding basic set of cluster configurations can be obtained from vector $\mathbf{r}_1$ and its basic set of cluster configurations by using the $^\star$ transformation. In the last column, the "free" coefficient values that minimize the energies of the corresponding configurations in the basic ray are presented.

Let us consider as an example the ray $\mathbf{r}_1$ for which $J_{01} < 0$ (ferromagnetic coupling) and all the other parameters are equal to zero. If $\alpha_1 = 0$, then, in this ray, the following triangular plaquette configurations have the lowest energies: °°, •°, °•, and •• for the types $a$ and $c$ of triangular plaquettes and all the possible configurations for the types $b$ and $d$. Configurations °• and •° have higher energies; configurations °• and °•, despite having the lowest energies, are incompatible with other configurations. Any global configuration, constructed with these local ones, is a ground-state configuration in this ray, that is, any global configuration, where local configurations °•, •°, °•, and °• are excluded, is a ground-state in this ray (and vice versa). It is clear that for these ground states all the red chains are ferromagnetic while the blue chains could be arbitrary. We refer to a structure constructed with a set of triangular plaquette configurations in such a way, that is, without any additional condition, as a "triangular" structure.

It should be noted that, at this stage, it is not yet proven that the $\mathbf{r}_1$ is a basic ray. As will become apparent below, the 22 rays listed in Table 1 are indeed basic rays, but they do not form a complete set.

## 2.4 Fully dimensional "triangular" phases

Although the set of basic rays is incomplete, many fully dimensional global ground-state configurations can be found using these basic rays, they are given in Table 2 and Figs. 3-7.

The first column of Table 2 gives the label for the regions in parameter space. In the second column, the triangular plaquette configurations that generate all the ground-state structures in this region are shown. Under the plaquette configurations basic rays (not all in most cases) for the region considered are also listed in this column. The third column lists some characteristics of the structure(s), such as energy (per six plaquettes), relative number of each plaquette configuration in the structure(s), and dimensionality of disorder. The coefficient in front of $h_1$ ($h_2$) in the expression for the energy, taken with opposite sign, is equal to the magnetization per one sublattice site of the "red" ("blue") sublattice. For region 2, for example, magnetization per site is 1 for the "red" sublattice and 1/3 for the "blue" sublattice. Among the basic vectors for every region, there are necessarily seven linearly independent ones, except for region 15. For this region seven basic vectors are determined but only six of them are linearly independent. However, one can prove that phase 15 is fully dimensional. To obtain all the basic vectors for this phase, larger clusters should be considered.

Table 1: Basic rays and basic sets of configurations for the Ising model on a honeycomb zigzag-ladder lattice.

| Basic ray $(J_{01}, J_{02}, J_{11}, J_{12}, J_2, h_1, h_2)$ | | Basic set of configurations $\mathbf{R}_i$ | "Free" coefficients |
|---|---|---|---|
| $\mathbf{r}_1$ | $(-1,0,0,0,0,0,0)$ | | $\alpha_1 = 0$ |
| $\mathbf{r}_1^\star$ | $(0,-1,0,0,0,0,0)$ | | $\alpha_2 = 0$ |
| $\mathbf{r}_2$ | $(1,0,-2,0,0,0,0)$ | | $\alpha_1 = 0$ |
| $\mathbf{r}_2^\star$ | $(0,1,0,-2,0,0,0)$ | | $\alpha_2 = 0$ |
| $\mathbf{r}_3$ | $(1,0,2,0,0,0,0)$ | | $\alpha_1 = 0$ |
| $\mathbf{r}_3^\star$ | $(0,1,0,2,0,0,0)$ | | $\alpha_2 = 0$ |
| $\mathbf{r}_4$ | $(2,0,0,0,1,0,0)$ | | $\alpha_1 = 1, \beta = 0$ |
| $\mathbf{r}_4^{\sim}$ | $(2,0,0,0,-1,0,0)$ | | $\alpha_1 = 1, \beta = 0$ |
| $\mathbf{r}_4^\star$ | $(0,2,0,0,1,0,0)$ | | $\alpha_2 = 1, \beta = 1$ |
| $\mathbf{r}_4^{\sim\star}$ | $(0,2,0,0,-1,0,0)$ | | $\alpha_2 = 1, \beta = 1$ |
| $\mathbf{r}_5$ | $(1,0,0,0,0,2,0)$ | | $\alpha_1 = 0, \gamma_1 = 1, \eta_1 = 0$ |
| $\mathbf{r}_5^-$ | $(1,0,0,0,0,-2,0)$ | | $\alpha_1 = 0, \gamma_1 = 1, \eta_1 = 0$ |
| $\mathbf{r}_5^\star$ | $(0,1,0,0,0,0,2)$ | | $\alpha_2 = 0, \gamma_2 = 1, \eta_2 = 0$ |
| $\mathbf{r}_5^{\star-}$ | $(0,1,0,0,0,0,-2)$ | | $\alpha_2 = 0, \gamma_2 = 1, \eta_2 = 0$ |
| $\mathbf{r}_6$ | $(0,0,1,0,0,2,0)$ | | $\gamma_1 = \tfrac{1}{2}, \delta_1 = 0, \eta_1 = 0$ |
| $\mathbf{r}_6^-$ | $(0,0,1,0,0,-2,0)$ | | $\gamma_1 = \tfrac{1}{2}, \delta_1 = 0, \eta_1 = 0$ |
| $\mathbf{r}_6^\star$ | $(0,0,0,1,0,0,2)$ | | $\gamma_2 = \tfrac{1}{2}, \delta_2 = 0, \eta_2 = 0$ |
| $\mathbf{r}_6^{\star-}$ | $(0,0,0,1,0,0,-2)$ | | $\gamma_2 = \tfrac{1}{2}, \delta_2 = 0, \eta_2 = 0$ |
| $\mathbf{r}_7$ | $(0,0,0,0,1,4,4)$ | | $\beta = 0, \gamma_1 = 1, \gamma_2 = 0, \delta_2 = 1, \eta_1 = 1$ |
| $\mathbf{r}_7^-$ | $(0,0,0,0,1,-4,-4)$ | | $\beta = 0, \gamma_1 = 1, \gamma_2 = 0, \delta_2 = 1, \eta_1 = 1$ |
| $\mathbf{r}_7^{\sim}$ | $(0,0,0,0,-1,4,-4)$ | | $\beta = 0, \gamma_1 = 1, \gamma_2 = 0, \delta_2 = 1, \eta_1 = 1$ |
| $\mathbf{r}_7^{\sim-}$ | $(0,0,0,0,-1,-4,4)$ | | $\beta = 0, \gamma_1 = 1, \gamma_2 = 0, \delta_2 = 1, \eta_1 = 1$ |

Let us consider an example. Structures 3 (see Fig. 3), composed of red *uu* chains and blue *ud* chains, are generated with triangular plaquette configurations, that is, every triangular plaquette in these structures should be one of these. This set of triangular plaquette configurations is a subset of basic sets $\mathbf{R}_i$ for basic rays $\mathbf{r}_1, \mathbf{r}_2, \mathbf{r}_2^\star, \mathbf{r}_3^\star, \mathbf{r}_4^\star, \mathbf{r}_4^{\sim\star}, \mathbf{r}_5, \mathbf{r}_5^\star, \mathbf{r}_5^{\star-}, \mathbf{r}_6, \mathbf{r}_7,$ and $\mathbf{r}_7^{\sim}$ (see Table 1). It means that, in the conic hull of this set of vectors, structures 3 are the ground-state ones. To calculate the energy of structures 3, it is sufficient to determine the relative number of each plaquette configuration in these structures (see Appendix A). These numbers are 2, 1, 1, 2, 2, and 4, respectively.

In Table 2, we give only one representative per class of structures. Other structures of the class can be obtained from the given one by applying three transformations ($\star$, $\sim$, and $^-$). For instance, the class of structures 5 contains eight structures (see Table 3 and Fig. 8): 5, $5^\star$,

Table 2: Fully dimensional regions and "triangular" ground-state structures of the Ising model on the honeycomb zigzag-ladder lattice. The structures (from 1 to 14) are numbered in order of decreasing magnetization of the "red" sublattice.

| | Generating configurations and basic rays | Characteristics of the structure(s) (energy per six plaquettes) |
|---|---|---|
| 1 | $\mathbf{r}_1, \mathbf{r}_1^\star, \mathbf{r}_2, \mathbf{r}_2^\star, \mathbf{r}_4^\sim, \mathbf{r}_4^{\sim\star},$ $\mathbf{r}_5, \mathbf{r}_5^\star, \mathbf{r}_6, \mathbf{r}_6^\star, \mathbf{r}_7, \mathbf{r}_7^\sim, \mathbf{r}_7^{\sim--}$ | $J_{01} + J_{02} + J_{11} + J_{12} + 4J_2 - h_1 - h_2$ $[\,1 \parallel 1 \parallel 2 \parallel 2\,]$, order |
| 2 | $\mathbf{r}_1, \mathbf{r}_2, \mathbf{r}_3^\star, \mathbf{r}_4^{\sim\star}, \mathbf{r}_5, \mathbf{r}_5^\star, \mathbf{r}_6, \mathbf{r}_6^\star, \mathbf{r}_7, \mathbf{r}_7^\sim$ | $\frac{1}{3}(3J_{01} - J_{02} + 3J_{11} - J_{12} + 4J_2 - 3h_1 - h_2)$ $[\,3 \parallel 1,2 \parallel 2,4 \parallel 4,2\,]$, 2D disorder |
| 3 | $\mathbf{r}_1, \mathbf{r}_2, \mathbf{r}_2^\star, \mathbf{r}_3^\star, \mathbf{r}_4, \mathbf{r}_4^{\sim\star},$ $\mathbf{r}_5, \mathbf{r}_5^\star, \mathbf{r}_5^{\star-}, \mathbf{r}_6, \mathbf{r}_7, \mathbf{r}_7^\sim$ | $J_{01} - J_{02} + J_{11} - h_1$ $[\,2 \parallel 1,1 \parallel 2,2 \parallel 4\,]$, 2D disorder |
| 4 | $\mathbf{r}_1, \mathbf{r}_1^\star, \mathbf{r}_2, \mathbf{r}_3^\star, \mathbf{r}_5, \mathbf{r}_6, \mathbf{r}_6^\star, \mathbf{r}_6^{\star-}, \mathbf{r}_7, \mathbf{r}_7^\sim$ | $J_{01} + J_{02} + J_{11} - J_{12} - h_1$ $[\,2 \parallel 1,1 \parallel 2,2 \parallel 2,2\,]$, 2D disorder |
| 5 | $\mathbf{r}_3^\star, \mathbf{r}_4^\sim, \mathbf{r}_4^{\sim\star}, \mathbf{r}_5, \mathbf{r}_6, \mathbf{r}_6^\star, \mathbf{r}_7^\sim$ | $\frac{1}{5}(J_{01} + J_{02} + J_{11} - 3J_{12} + 12J_2 - 3h_1 - h_2)$ $[\,1,2,2 \parallel 1,2,2 \parallel 4,6 \parallel 2,4,4\,]$, order |
| 6 | $\mathbf{r}_1^\star, \mathbf{r}_3^\star, \mathbf{r}_4, \mathbf{r}_5, \mathbf{r}_6, \mathbf{r}_6^\star, \mathbf{r}_6^{\star-}, \mathbf{r}_7$ | $\frac{1}{2}(2J_{02} - 2J_{12} - 4J_2 - h_1)$ $[\,1,2,1 \parallel 2,2 \parallel 4,4 \parallel 4,2,2\,]$, 2D disorder |
| 7 | $\mathbf{r}_2, \mathbf{r}_2^\star, \mathbf{r}_4, \mathbf{r}_4^\star, \mathbf{r}_5, \mathbf{r}_5^\star, \mathbf{r}_7$ | $\frac{1}{3}(-J_{01} - J_{02} + J_{11} + J_{12} - 4J_2 - h_1 - h_2)$ $[\,1,1,1 \parallel 1,1,1 \parallel 2,4 \parallel 2,4\,]$, 1D disorder |
| 8 | $\mathbf{r}_3, \mathbf{r}_3^\star, \mathbf{r}_4, \mathbf{r}_4^\star, \mathbf{r}_5, \mathbf{r}_5^\star, \mathbf{r}_6, \mathbf{r}_6^\star, \mathbf{r}_7$ | $\frac{1}{3}(-J_{01} - J_{02} - J_{11} - J_{12} - 4J_2 - h_1 - h_2)$ $[\,1,2 \parallel 1,2 \parallel 2,4 \parallel 2,4\,]$, order |
| 9 | $\mathbf{r}_3, \mathbf{r}_3^\star, \mathbf{r}_4^\sim, \mathbf{r}_4^{\sim\star}, \mathbf{r}_5, \mathbf{r}_5^\star, \mathbf{r}_6, \mathbf{r}_6^\star$ | $\frac{1}{3}(-J_{01} - J_{02} - J_{11} - J_{12} + 4J_2 - h_1 - h_2)$ $[\,1,2 \parallel 1,2 \parallel 2,2,2 \parallel 2,2,2\,]$, 2D disorder |
| 10 | $\mathbf{r}_2^\star, \mathbf{r}_3, \mathbf{r}_4^\sim, \mathbf{r}_4^{\sim\star}, \mathbf{r}_5, \mathbf{r}_5^\star, \mathbf{r}_6$ | $\frac{1}{3}(-J_{01} - J_{02} - J_{11} + J_{12} + 4J_2 - h_1 - h_2)$ $[\,1,2 \parallel 1,1,1 \parallel 2,2,2 \parallel 2,2,2\,]$, 3D disorder |
| 11 | $\mathbf{r}_2^\star, \mathbf{r}_3, \mathbf{r}_3^\star, \mathbf{r}_4, \mathbf{r}_4^{\sim\star}, \mathbf{r}_5, \mathbf{r}_5^\star, \mathbf{r}_5^{\star-}, \mathbf{r}_6$ | $\frac{1}{3}(-J_{01} - 3J_{02} - J_{11} - h_1)$ $[\,2,4 \parallel 3,3 \parallel 4,2,4,2 \parallel 4,8\,]$, 2D disorder |
| 12 | $\mathbf{r}_1, \mathbf{r}_1^\star, \mathbf{r}_3, \mathbf{r}_3^\star, \mathbf{r}_4, \mathbf{r}_4^\star,$ $\mathbf{r}_6, \mathbf{r}_6^-, \mathbf{r}_6^\star, \mathbf{r}_6^{\star-}, \mathbf{r}_7, \mathbf{r}_7^-$ | $J_{01} + J_{02} - J_{11} - J_{12} - 4J_2$ $[\,1,1 \parallel 1,1 \parallel 2,2 \parallel 2,2\,]$, order |
| 13 | $\mathbf{r}_1^\star, \mathbf{r}_2, \mathbf{r}_3, \mathbf{r}_3^\star, \mathbf{r}_4, \mathbf{r}_4^\sim, \mathbf{r}_5, \mathbf{r}_5^-, \mathbf{r}_6^\star, \mathbf{r}_6^{\star-}$ | $-J_{01} + J_{02} - J_{12}$ $[\,1,1 \parallel 1,1 \parallel 2,2 \parallel 1,1,1,1\,]$, 2D disorder |
| 14 | $\mathbf{r}_2, \mathbf{r}_2^\star, \mathbf{r}_3, \mathbf{r}_3^\star,$ $\mathbf{r}_4, \mathbf{r}_4^\sim, \mathbf{r}_4^\star, \mathbf{r}_4^{\sim\star}, \mathbf{r}_5, \mathbf{r}_5^-, \mathbf{r}_5^\star, \mathbf{r}_5^{\star-}$ | $-J_{01} - J_{02}$ $[\,1,1 \parallel 1,1 \parallel 2,2 \parallel 2,2\,]$, 2D disorder |
| 15 | $\mathbf{r}_1^\star, \mathbf{r}_3, \mathbf{r}_3^\star, \mathbf{r}_5, \mathbf{r}_6, \mathbf{r}_6^\star, \mathbf{r}_6^{\star-}$ | $\frac{1}{3}(-J_{01} + 3J_{02} - J_{11} - 3J_{12} - h_1)$ $[\,2,4 \parallel 3,3 \parallel 4,2,4,2 \parallel 2,4,2,4\,]$, disorder |



Figure 3: Phases 2, 3, 4, 5, 6, 8, 11, 12, 13, 14, and 15. For each phase, the left hand panel shows the configuration of the spins within each chain of a hexagonal tube (its development is depicted). The larger colored circles at the bottom of the chains are a key that indicate how the spins along a chain are distributed in each of the arrangements shown in the right hand panels. For each phase, the right hand panel shows the setting of the spins viewed down the chains from above. The configurations of triangular plaquettes are also given. Phases 5, 8, and 12 are ordered, while phases 2, 3, 4, 6, 11, 13, 14, and 15 are disordered (the disorder is two-dimensional).

$\widetilde{5}$, $\widetilde{5}^\star$, $\overline{5}$, $\overline{5^\star}$, $\overline{\widetilde{5}}$, and $\overline{\widetilde{5}^\star}$. This is the maximum number of structures in one class. It should be noted here that $\overline{\widetilde{n^*}} = (\tilde{n})^*$. The class of structures 14 contains only structure 14 because this structure is symmetric with respect to all the three transformations. There are 69 "triangular" phases in total.

It should be also noted that phases 5, 6, 7, and 10 are not possible if only uncoupled "red" and "blue" zigzag-ladders are considered, i.e. if $J_2 = 0$ (see Ref. [33] for ground states of one zigzag-ladder).

Table 3: Structure 5 and seven other structures obtained from it using transformations $^\star, \widetilde{\phantom{x}}$, and $^-$.

| | Generating configurations and basic rays | Characteristics of the structure (energy per six plaquettes) |
|---|---|---|
| 5 | $\mathbf{r}_3^\star, \mathbf{r}_{\widetilde{4}}, \mathbf{r}_{\widetilde{4}}^{\star}, \mathbf{r}_5, \mathbf{r}_6, \mathbf{r}_6^\star, \mathbf{r}_{\widetilde{7}}$ | $\frac{1}{5}(J_{01} + J_{02} + J_{11} - 3J_{12} + 12J_2 - 3h_1 - h_2)$ <br> $[\, 1,2,2 \parallel 1,2,2 \parallel 4,6 \parallel 2,4,4 \,]$, order |
| $5^\star$ | $\mathbf{r}_3, \mathbf{r}_{\widetilde{4}}, \mathbf{r}_{\widetilde{4}}^{\star}, \mathbf{r}_5^\star, \mathbf{r}_6, \mathbf{r}_6^\star, \mathbf{r}_{\widetilde{7}}^{-}$ | $\frac{1}{5}(J_{01} + J_{02} - 3J_{11} + J_{12} + 12J_2 - h_1 - 3h_2)$ <br> $[\, 1,2,2 \parallel 1,2,2 \parallel 2,4,4 \parallel 4,6 \,]$, order |
| $\widetilde{5}$ | $\mathbf{r}_3^\star, \mathbf{r}_4, \mathbf{r}_4^\star, \mathbf{r}_5, \mathbf{r}_6, \mathbf{r}_6^{\star-}, \mathbf{r}_7$ | $\frac{1}{5}(J_{01} + J_{02} + J_{11} - 3J_{12} - 12J_2 - 3h_1 + h_2)$ <br> $[\, 1,2,2 \parallel 2,2,1 \parallel 6,4 \parallel 4,2,4 \,]$, order |
| $\widetilde{5}^\star$ | $\mathbf{r}_3, \mathbf{r}_4, \mathbf{r}_4^\star, \mathbf{r}_5^\star, \mathbf{r}_6^-, \mathbf{r}_6^\star, \mathbf{r}_7$ | $\frac{1}{5}(J_{01} + J_{02} - 3J_{11} + J_{12} - 12J_2 + h_1 - 3h_2)$ <br> $[\, 2,2,1 \parallel 1,2,2 \parallel 4,2,4 \parallel 6,4 \,]$, order |
| $\overline{5}$ | $\mathbf{r}_3^\star, \mathbf{r}_{\widetilde{4}}, \mathbf{r}_{\widetilde{4}}^{\star}, \mathbf{r}_5^-, \mathbf{r}_6^-, \mathbf{r}_6^{\star-}, \mathbf{r}_{\widetilde{7}}^{-}$ | $\frac{1}{5}(J_{01} + J_{02} + J_{11} - 3J_{12} + 12J_2 + 3h_1 + h_2)$ <br> $[\, 2,2,1 \parallel 2,2,1 \parallel 6,4 \parallel 4,4,2 \,]$, order |
| $\overline{5^\star}$ | $\mathbf{r}_3, \mathbf{r}_{\widetilde{4}}, \mathbf{r}_{\widetilde{4}}^{\star}, \mathbf{r}_5^{\star-}, \mathbf{r}_6^-, \mathbf{r}_6^{\star-}, \mathbf{r}_{\widetilde{7}}$ | $\frac{1}{5}(J_{01} + J_{02} - 3J_{11} + J_{12} + 12J_2 + h_1 + 3h_2)$ <br> $[\, 2,2,1 \parallel 2,2,1 \parallel 4,4,2 \parallel 6,4 \,]$, order |
| $\overline{\widetilde{5}}$ | $\mathbf{r}_3^\star, \mathbf{r}_4, \mathbf{r}_4^\star, \mathbf{r}_5^-, \mathbf{r}_6^-, \mathbf{r}_6^\star, \mathbf{r}_7^-$ | $\frac{1}{5}(J_{01} + J_{02} + J_{11} - 3J_{12} - 12J_2 + 3h_1 - h_2)$ <br> $[\, 2,2,1 \parallel 1,2,2 \parallel 4,6 \parallel 4,2,4 \,]$, order |
| $\overline{\widetilde{5}^\star}$ | $\mathbf{r}_3, \mathbf{r}_4, \mathbf{r}_4^\star, \mathbf{r}_5^{\star-}, \mathbf{r}_6, \mathbf{r}_6^{\star-}, \mathbf{r}_7^-$ | $\frac{1}{5}(J_{01} + J_{02} - 3J_{11} + J_{12} - 12J_2 - h_1 + 3h_2)$ <br> $[\, 1,2,2 \parallel 2,2,1 \parallel 4,2,4 \parallel 4,6 \,]$, order |

## 2.5 Disorder (degeneracy) of the phases

Let us analyze the disorder of phases next. Phases 1, 5, 8, and 12 (and the phases obtained from them by using the transformations described above) are ordered. All the other phases are disordered, that is, there are an infinite number of structures with the same energy. A disorder can be characterized by its dimensionality. For instance, the disorder of phase 7 is

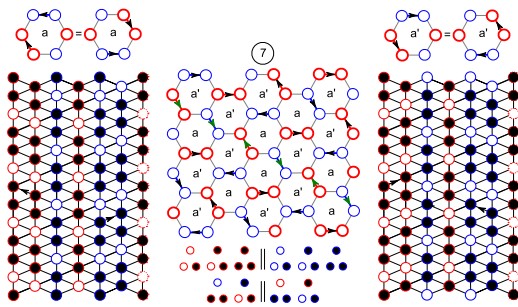

Figure 4: Phase 7. The structures of this phase are constructed with the two configurations of hexagonal tube: $a$ and $a'$. The arrows show the directions of shift for chains. The global arrow configuration (constructed with two hexagonal configurations) is fully determined by a line of arrows (depicted in olive). Therefore, the disorder is one-dimensional.

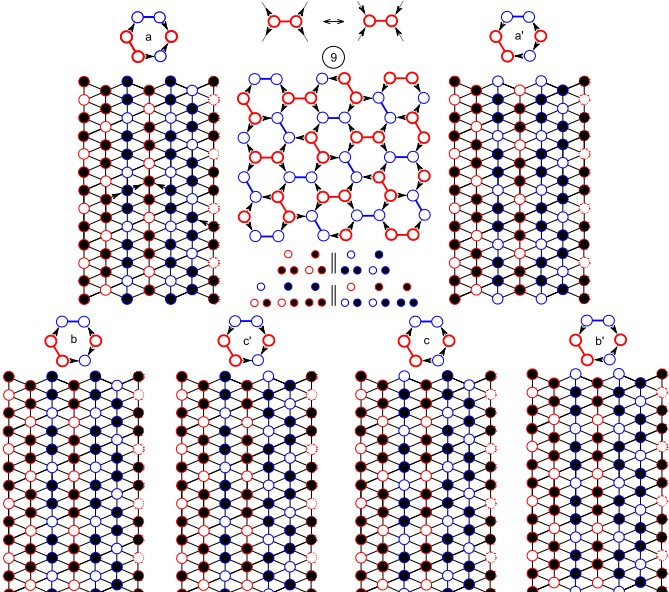

Figure 5: Structures 9 are composed of six hexagonal tube configurations corresponding to six arrow configurations of hexagons ($a$, $b$, $c$, $a'$, $b'$, and $c'$) in which two arrows are pointing clockwise and two others anticlockwise or vice versa. The global arrow configuration generated by local configuration $b$ is depicted. The substitution shown in the upper part of the figure can be made locally without violating the ground state rules. Therefore, the disorder is two-dimensional.

one-dimensional, as shown in Fig. 4. The structures of this phase are constructed with two hexagonal tube configurations that can be depicted as hexagons with two arrows showing the shift of the corresponding chains. The global lattice configuration is mapped on two-dimensional arrow configurations. It is easy to see, such an arrow configuration is determined by an arbitrary one-dimensional sequence of arrows. So, the disorder is one-dimensional.

The disorder of phase 2 is two-dimensional, since all the chains are ordered but every up-up-down ($uud$) chain can be in three different positions. That is, there is a perfect order along the $c$ direction (along the chains) but a disorder in the $ab$-plane. The disorder of phase 4 is also two-dimensional, since every "blue" ladder can be in two different positions. It is shown in Fig. 5 that the disorder of phase 9 is two-dimensional as well, because all the chains are ordered and, in the structure generated by arrow configuration $b$, the local arrow configurations depicted in the upper part of the figure are interchangeable.

The disorder of phase 10 is three-dimensional, that is, the degeneracy is macroscopic and therefore there exists a residual entropy in this phase. Let us prove this. The structures of phase 10 are constructed with ten hexagonal tube configurations, shown in Fig. 6. These tube configurations are composed of identical chains, $uud$. The shift of a chain configuration when passing to a neighboring one can be indicated by an arrow. So, we have ten arrow configurations of hexagons. Notation $x'$ means that all the arrows in the hexagon are opposite to those in hexagon configuration $x$. In Fig. 7, an example of arrow configurations and another representation of the same structures, by explicit indication of chain positions, are given. For the "shaded" chains the shifts of all the three neighboring chains are equal. In such blue chains, one of two spins in each oval (Fig. 7, right panel) can point in an arbitrary direction with the other being opposite. So, the disorder in phase 10 is three-dimensional.

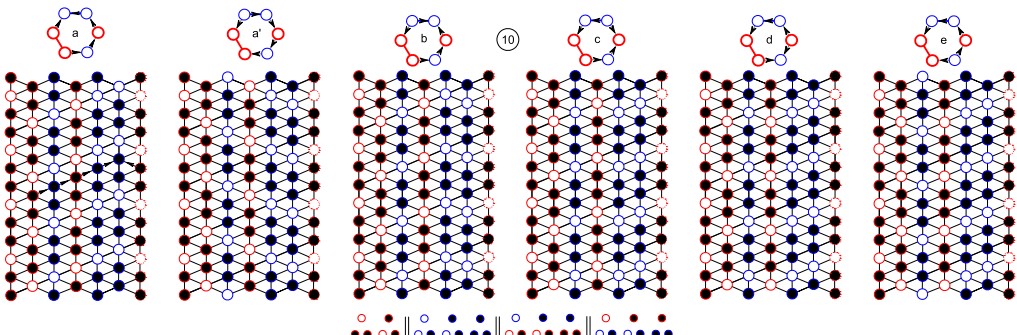

Figure 6: Structures 10 are composed of ten hexagonal tube configurations corresponding to ten arrow configurations of hexagons, $a$, $b$, $c$, $d$, $e$, $a'$, $b'$, $c'$, $d'$, and $e'$ (the last four are not shown), in which one arrow is pointing clockwise and four others anticlockwise or vice versa.

## 2.6 Fully dimensional "nontriangular" phases

The sets of basic vectors are complete for only four (4, 11, 13, and 14) of the fully-dimensional regions stated in Table 2. The proof of the completeness is given in Appendix B which lists the 6-faces for these regions. But even if the set of basic vectors for a "triangular" phase region is incomplete, the majority of 6-faces of the polyhedral cone generated by this set are 6-faces of this region. Then, even if the neighboring phase is not a "triangular" one, it is possible to determine the ground-state structure(s) for this phase. Such structures should have a maximum number of new triangular configurations which are absent in structures of the "triangular" phase but present in ground-state structures at the common boundary (6-face). We found nine phases (more exactly, nine classes) of this type. The list of these phases is given in Table 4 (one representative per class) and the corresponding structures are depicted in Figs. 9-13. In these figures, new triangular configurations are framed by dotted squares. It should be noted that the sets of triangular configurations in Table 4 and in the figures are the sets of ground-state triangular configurations for six-dimensional boundaries. As one can see from Figs. 9-11, some "nontriangular" structures, in contrast to "triangular" ones, are composed of two different types of red ladder (phases 16, 18, 19, 22, and 23) or blue ladder (phases 20 and 21) configurations. These phases are due to the interaction between red and blue ladders, they are therefore excluded from considerations in the one-dimensional models, such as 1D ANNNI model used in Ref. [34].

It is worthwhile studying the disorder of these "nontriangular" phases. Phases 17 and 20 are ordered. As it is clear from Fig. 9 (upper panel), the disorder of phase 16 is one-dimensional, since the structure is completely determined by a sequence of arrows showing the shifts of neighboring red $uud$ chains. The disorder of phases 18, 21, 22, and 23 is two-dimensional due to $ud$ chains. A complex disorder is present in phase 24. The structures of this phase can be mapped on two-dimensional arrow configurations composed of ten hexagon arrow configurations in which one arrow is aligned clockwise and the five others anticlockwise or vice versa, the arrow between blue sites being aligned with the majority of the arrows. Three arrows depicted in Fig. 13 (left hand panel) produce an infinite half-chain of hexagon arrow configurations. At the first sight, the local arrow configuration shown in Fig. 13 (middle panel) should produce a three-dimensional disorder. However, the number of this arrow configurations is infinitesimal, since every configuration of this type generates at least two half-chains of hexagons. So, the disorder is not three-dimensional but two- or, possibly, even one-dimensional.

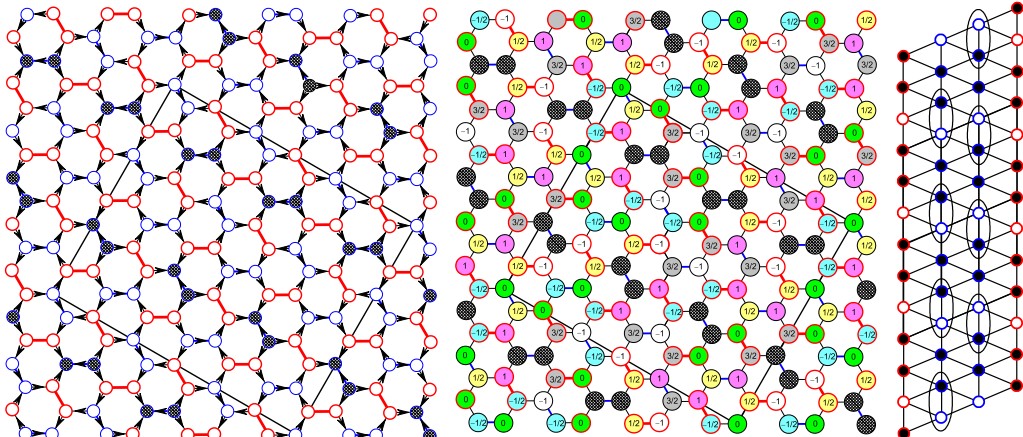

Figure 7: One of possible structures of phase 10 is shown in two ways, (left panel) with the help of arrow configurations and (middle panel) by indicating the shift of each chain, in the units of the in-chain spin distance. The unit cell is also indicated. For the "shaded" chains the shifts of all the three neighboring chains are equal. In such blue chains (right panel), one of two spins in each oval can be arbitrary, the other being of opposite direction. The disorder in phase 10 is three-dimensional, i.e. this phase is macroscopically degenerate.

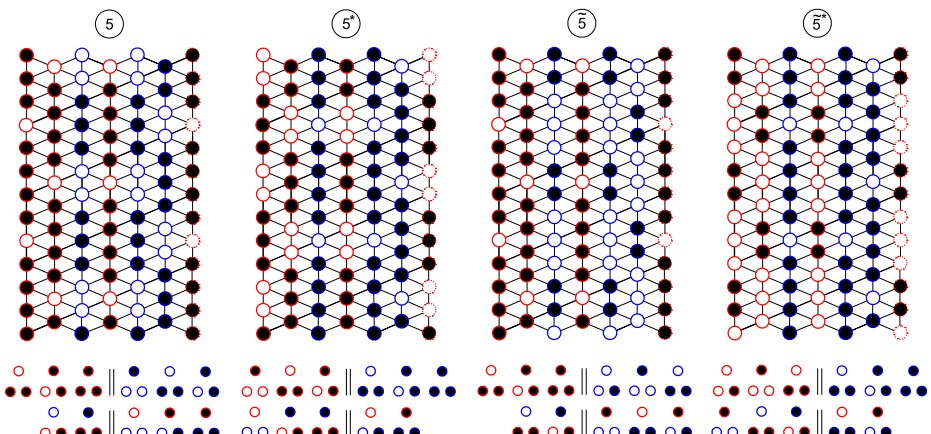

Figure 8: Structure 5 and three related structures obtained from it by using transformations $^\star$, and $^\sim$. Transformation $^-$ (spin flip on both sublattices) gives additional four structures. Only one hexagonal tube is shown for each structure.

## 2.7 Ground-state phase diagrams in the $(h_1, h_2)$-plane

Consider the ground-state phase diagrams in the $(h_1, h_2)$-plane. Although the solution of the ground-state problem is incomplete, at some particular values of the parameters $J_{01}$, $J_{02}$, $J_{11}$, $J_{12}$, and $J_2$, it is possible to construct exact and complete phase diagrams. Six examples of such diagrams are given in Fig. 14. The boundaries shown with dotted lines are not strictly proven.

When both the interaction parameter values and the external fields, $h_1$ and $h_2$, are fixed we have a single point on the phase diagram. By continuously varying the value (and possibly the direction) of an external field, instead of a point, we generate a line of transitions on the phase diagram. For instance, increasing magnetic field along the $a$ axis in SrEr$_2$O$_4$ that corresponds

Figure 9: Phases 16, 17, 18, 19, 20, 22, and 23. They appear at boundary of phases 2, 3, 4, 4, 10, 13, and 13, respectively. The triangular configurations shown below the structures are the ground-state configurations at these boundaries. New triangular configurations are surrounded by dotted squares. The principle of these structures construction is to find at the given boundary the structures containing maximum number of such configurations. To show chains, only one hexagonal tube configuration is depicted for each phase. A more detailed picture of phase 20 is shown in Fig. 7 (middle panel) with "dashed" $ddu$ chains.

Table 4: Fully dimensional regions and "nontriangular" ground-state structures of the Ising model on the honeycomb zigzag-ladder lattice.

| Boundary | Triangular configurations and basic rays | Characteristics of "nontriangular" structures |
|---|---|---|
| (2, 16) | $\mathbf{r}_2, \mathbf{r}_3^\star, \mathbf{r}_5, \mathbf{r}_5^\star, \mathbf{r}_6^\star, \mathbf{r}_7$ | $\frac{1}{3}(J_{01} - J_{02} + 2J_{11} - J_{12} - 2h_1 - h_2)$ <br> [ 1, 1, 4 ∥ 2, 4 ∥ 4, 4, 4 ∥ 2, 8, 2 ],  1D disorder |
| (3, 17) | $\mathbf{r}_2^\star, \mathbf{r}_4^\star, \mathbf{r}_5, \mathbf{r}_5^\star, \mathbf{r}_6^\star, \mathbf{r}_7$ | $\frac{1}{5}(J_{01} - 3J_{02} + J_{11} + J_{12} - 4J_2 - 3h_1 - h_2)$ <br> [ 1, 2, 2 ∥ 2, 2, 1 ∥ 4, 4, 2 ∥ 2, 8 ],  order |
| (4, 18) | $\mathbf{r}_1^\star, \mathbf{r}_2, \mathbf{r}_3^\star, \mathbf{r}_5, \mathbf{r}_6^\star, \mathbf{r}_6^{\star-}, \mathbf{r}_7$ | $\frac{1}{4}(2J_{01} + 4J_{02} + 3J_{11} - 4J_{12} - 4J_2 - 3h_1)$ <br> [ 1, 1, 6 ∥ 4, 4 ∥ 8, 4, 4 ∥ 8, 2, 6 ],  2D disorder |
| (4, 19) | $\mathbf{r}_1, \mathbf{r}_1^\star, \mathbf{r}_2, \mathbf{r}_3^\star, \mathbf{r}_6^\star, \mathbf{r}_6^{\star-}, \widetilde{\mathbf{r}_7}$ | $\frac{1}{2}(2J_{01} + 2J_{02} + 2J_{11} - 2J_{12} + 4J_2 - h_1)$ <br> [ 1, 3 ∥ 2, 2 ∥ 2, 2, 4 ∥ 2, 2, 4 ],  2D disorder |
| (10, 20) | $\mathbf{r}_2^\star, \mathbf{r}_3, \widetilde{\mathbf{r}_4}, \widetilde{\mathbf{r}_4}^{\star}, \mathbf{r}_5, \mathbf{r}_6$ | $\frac{1}{9}(-3J_{01} - 3J_{02} - 3J_{11} + 3J_{12} + 12J_2 - 3h_1 - h_2)$ <br> [ 3, 6 ∥ 1, 3, 3, 2 ∥ 8, 4, 6 ∥ 2, 4, 8, 4 ],  order |
| (11, 21) | $\mathbf{r}_2^\star, \mathbf{r}_3, \mathbf{r}_4^\star, \mathbf{r}_5, \mathbf{r}_5^\star, \mathbf{r}_6$ | $\frac{1}{9}(-3J_{01} - 7J_{02} - 3J_{11} + J_{12} - 4J_2 - 3h_1 - h_2)$ <br> [ 3, 6 ∥ 4, 4, 1 ∥ 4, 4, 8, 2 ∥ 4, 2, 12 ],  2D disorder |
| (13, 22) | $\mathbf{r}_1^\star, \mathbf{r}_2, \mathbf{r}_3^\star, \widetilde{\mathbf{r}_4}, \mathbf{r}_5, \mathbf{r}_6^\star, \mathbf{r}_6^{\star-}$ | $\frac{1}{4}(-2J_{01} + 4J_{02} + J_{11} - 4J_{12} + 4J_2 - h_1)$ <br> [ 3, 3, 2 ∥ 4, 4 ∥ 8, 4, 4 ∥ 4, 4, 2, 6 ],  2D disorder |
| (13, 23) | $\mathbf{r}_1^\star, \mathbf{r}_3, \mathbf{r}_3^\star, \widetilde{\mathbf{r}_4}, \mathbf{r}_5, \mathbf{r}_6^\star, \mathbf{r}_6^{\star-}$ | $\frac{1}{12}(-10J_{01} + 12J_{02} - J_{11} - 12J_{12} + 4J_2 - h_1)$ <br> [ 9, 2, 13 ∥ 12, 12 ∥ 24, 20, 4 ∥ 12, 12, 10, 14 ], <br> 2D disorder |
| (14, 24) | $\mathbf{r}_2^\star, \mathbf{r}_3, \mathbf{r}_4, \mathbf{r}_4^\star, \mathbf{r}_5, \mathbf{r}_5^\star$ | $\frac{1}{5}(-3J_{01} - 3J_{02} - J_{11} + J_{12} - 4J_2 - h_1 - h_2)$ <br> [ 1, 1, 3 ∥ 2, 2, 1 ∥ 2, 2, 6 ∥ 2, 2, 6 ],  disorder |

to a passage along the $h_1$ axis in Fig. 14 (left middle panel) we have the following sequence of the phases: 13, 23, 15, $\tilde{6}$ and 4 (see also Fig. 15).

Let us show how to prove that the point where three phases, for instance, 2, 5, and 9, meet, exists in a ground-state phase diagram. This point is determined by the following set of vectors (common for all the three phases), $\{\mathbf{r}_3^\star, \widetilde{\mathbf{r}_4}^{\star}, \mathbf{r}_5, \mathbf{r}_6, \mathbf{r}_6^\star\}$. At fixed $J_{01}, J_{02}, J_{11} \, J_{12}$, and $J_2$ the solution of the equation

$$a_3^\star \mathbf{r}_3^\star + a_4^{\widetilde{\star}} \widetilde{\mathbf{r}_4}^{\star} + a_5 \mathbf{r}_5 + a_6 \mathbf{r}_6 + a_6^\star \mathbf{r}_6^\star = (J_{01}, J_{02}, J_{11}, J_{12}, J_2, h_1, h_2) \qquad (4)$$

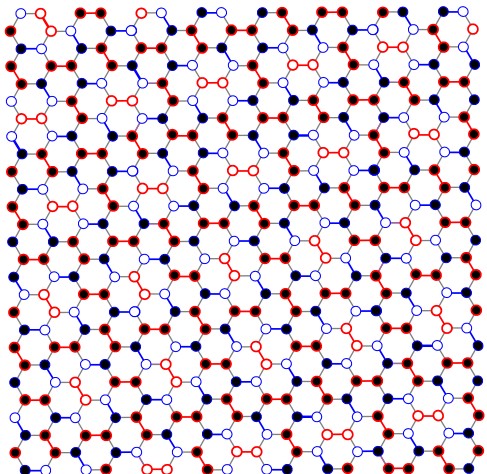

Figure 10: Disorder of phase 19. Open and filled circles denote two types of ferromagnetic chains. A similar disorder is present in phases 18, 22, and 23.



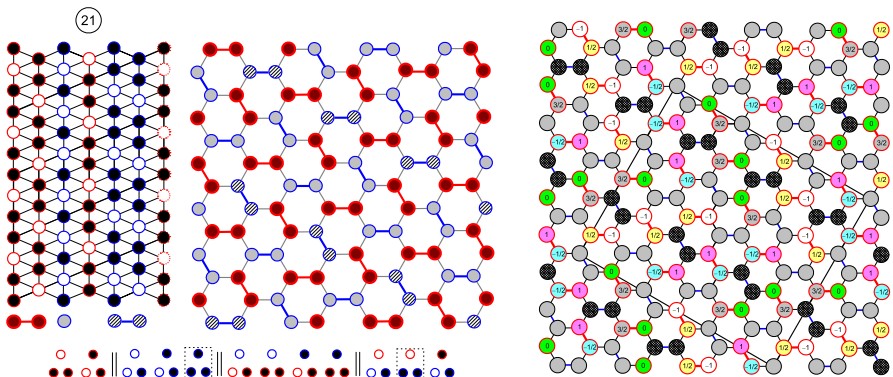

Figure 11: Phase 21. There is a two-dimensional disorder in this phase due to the presence of *ud* chains. Right panel gives a more detailed picture of the structure (compare with Fig. 7, middle panel).

is

$$a_3^\star = J_{02} + 2J_2, \; a_4^{\widetilde{\star}} = -J_2, \; a_5 = J_{01}, \; a_6 = J_{11}, \; a_6^\star = -2J_{02} + J_{12} - 4J_2,$$
$$h_1 = 2J_{01} + 2J_{11}, \; h_2 = -4J_{02} + 2J_{12} - 8J_2. \tag{5}$$

For $J_{01} = 1.0$, $J_{02} = 0.65$, $J_{11} = 0.60$, $J_{12} = 0.75$, and $J_2 = -0.25$ all the five coefficients are nonnegative, so, the linear combination in the left side of Eq. 4 belongs to the conical hull of the set of vectors, and, therefore, for these values of parameters, the point where the phases 2, 5, and 9 meet exists in the ground-state phase diagram for these values of parameters. It is the point $h_1 = 3.2$, $h_2 = 0.9$.

In a similar way one can, for instance, find conditions for the existence of the region 13 – region 23 boundary in the $(h_1, h_2)$-plane,

$$J_{11} > 0, \; J_{12} > 0, \; J_2 < 0, \; 2J_{01} - J_{11} + 4J_2 > 0, \; J_{12} - 2J_{02} > 0. \tag{6}$$

Then, for this boundary we have

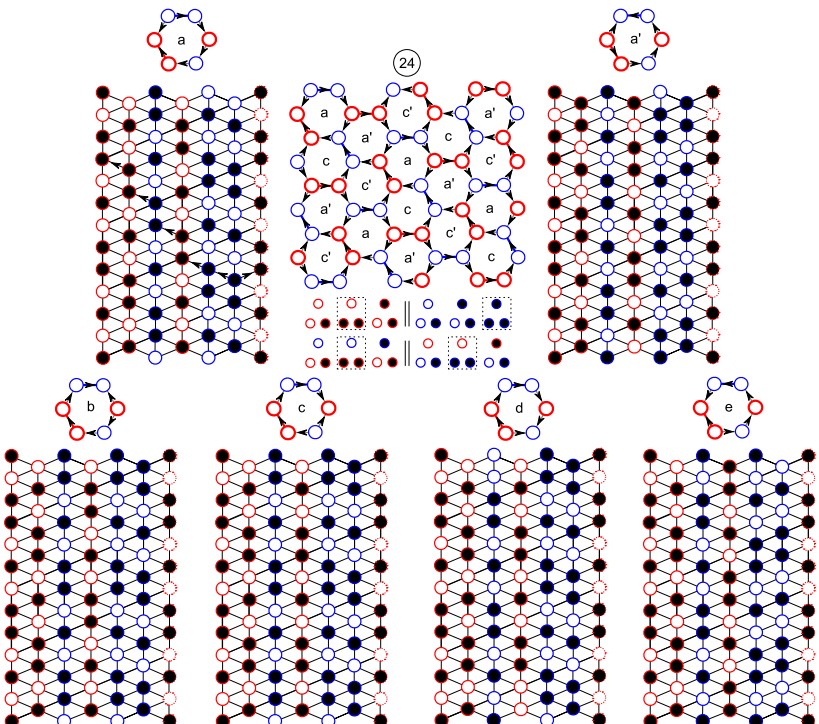

Figure 12: Structures 24 are determined by ten arrow configurations of hexagon ($a$, $b$, $c$, $d$, $e$, $a'$, $b'$, $c'$, $d'$, and $e'$) in which one arrow is pointing clockwise and five others anticlockwise or vice versa. The arrow between blue sites is aligned with the majority of the arrows. An example of global arrow configuration is also shown.

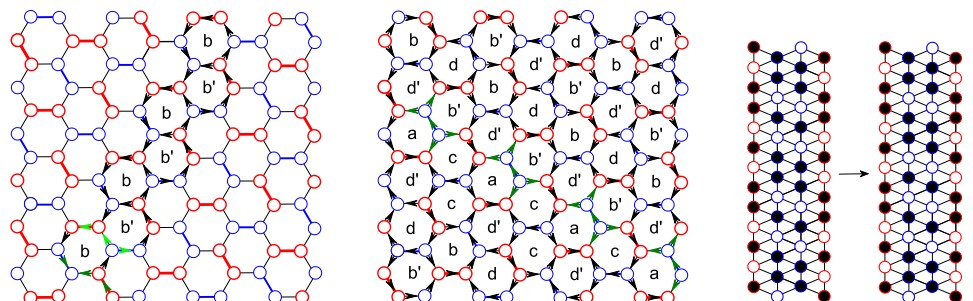

Figure 13: Disorder of phase 24. Half-chain of hexagon arrow configurations for phase 24 (see Fig. 12) is completely determined by the three arrow depicted in olive or in green (left hand panel). Local arrow configuration shown in olive (middle panel) could lead to a three-dimensional disorder because a rearrangement of spins in blue chains of such configuration is possible (right hand panel). However, the number of this arrow configurations is infinitely small, since every configuration of this type generates at least two half-chains of hexagons.

$$h_1 = 2J_{01} - J_{11} + 4J_2 \,,$$
$$\begin{cases} 4J_{02} - 2J_{12} < h_2 < 2J_{12} - 4J_{02} & \text{if } J_{02} > 0 \,, \\ -2J_{12} < h_2 < 2J_{12} & \text{if } J_{02} < 0 \,. \end{cases} \tag{7}$$

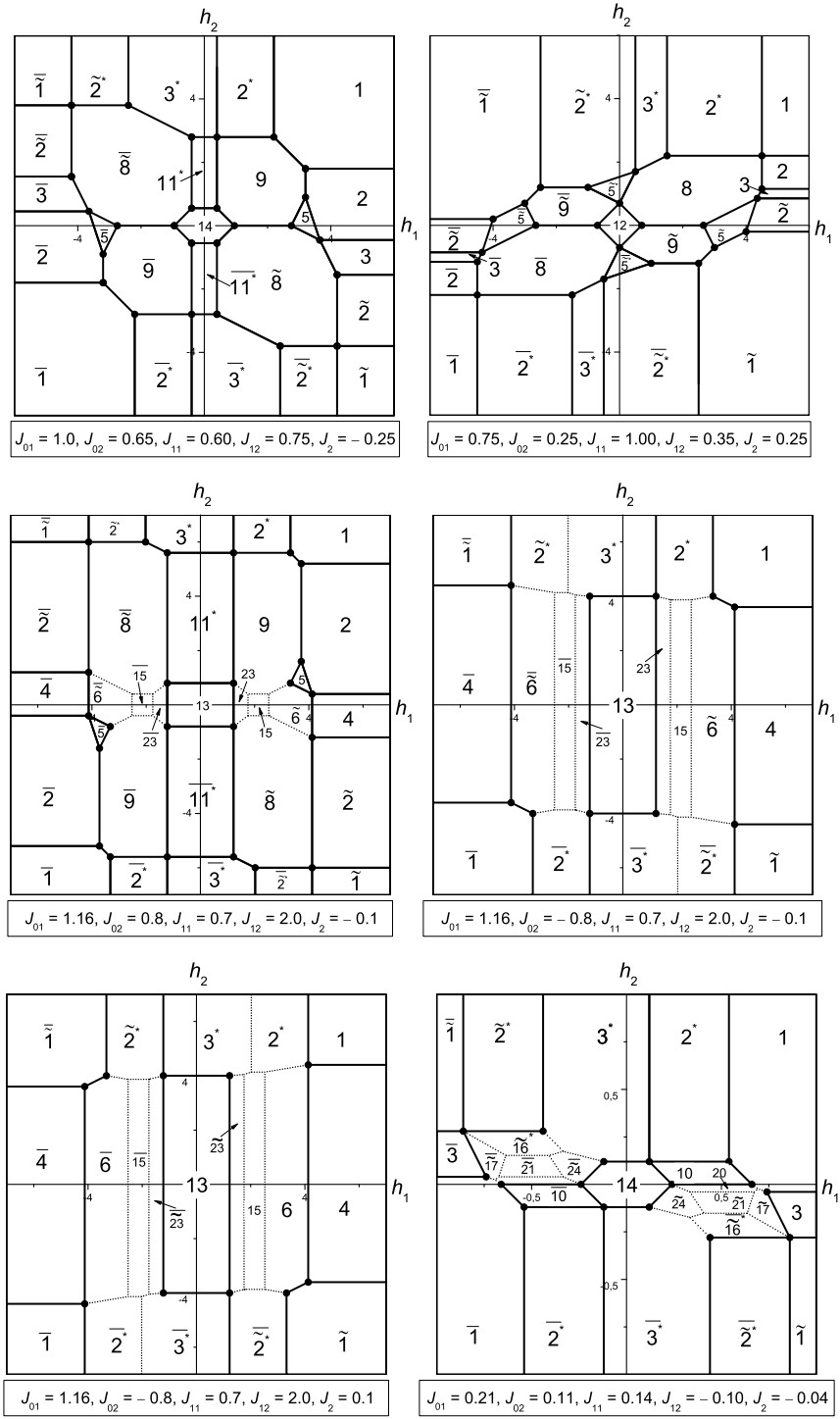

Figure 14: Examples of ground-state phase diagrams in the $(h_1, h_2)$-plane (the fields and couplings are shown in arbitrary units). Some diagrams are not completely proven, particularly the transitions depicted by the doted lines.

# 3 Application to SrRE$_2$O$_4$ and BaRE$_2$O$_4$ compounds

In this section, we consider an application of the theoretical approach discussed above to the magnetic properties of the two families of rare-earth compounds, SrRE$_2$O$_4$ and BaRE$_2$O$_4$. We start by briefly summarizing what is experimentally known about the ground state configurations of these zigzag-ladder magnets, particularly focusing on the in-field behaviour of SrEr$_2$O$_4$, SrHo$_2$O$_4$, SrDy$_2$O$_4$ and BaDy$_2$O$_4$.

The crystal structure of these compounds is very close to the one depicted in Fig. 1, with two RE ions in different positions forming a set of triangular ladders running along the $c$ axis [1]. The ladders are arranged in a honeycomb-like lattice in the $a$-$b$ plane, however, the honeycombs are significantly distorted so that the distances between the ions are not identical, which results in the need to introduce different exchange couplings, $J_{11} \neq J_{12} \neq J_2$ in our model.

One important question to address here is to what degree the SrRE$_2$O$_4$ and BaRE$_2$O$_4$ compounds could be characterized as Ising-type magnets. The answer to this question should come most naturally from considering the effects of crystal fields (CFs), however, the task of establishing the sets of relevant CF parameters for the two ions in crystallographically inequivalent positions is far from trivial. Because of the low overall symmetry and the large number of atoms in a unit cell, interpretation of inelastic neutron scattering data does not necessarily return a unique set of CF parameters unless supplemented by optical and electron paramagnetic resonance measurements, and so far this has only been done for SrEr$_2$O$_4$ [35]. For SrEr$_2$O$_4$, the observed largely anisotropic g factors for the Er$^{3+}$ ions in both crystallographically inequivalent sites [35] prove the applicability of the Ising model. For SrHo$_2$O$_4$ anf SrDy$_2$O$_4$, the results of the inelastic neutron scattering were also interpreted as consistent with the Ising chain model [36].

In zero field, the Er ions positioned in SrEr$_2$O$_4$ on different sites participate in the formation of two different magnetic systems acting almost independently of each other [37,38]. Er1 sites form a long-range antiferromagnetic order with the magnetic moments aligned parallel to the $c$ direction. For this site, each ladder is made of the two ferromagnetic chains aligned antiparallel to each other. Er2 sites participate in the formation of a short-range one-dimensional order, where the spins lay in the $a$-$b$ plane, and demonstrate very strong antiferromagnetic in-chain correlations (along the $c$ axis) with much weaker correlations between the chains (that is in the direction normal to the $c$ axis). In the absence of an external field, phase 13 is realized in SrEr$_2$O$_4$ (without degeneracy of the Er1 subsystem).

In SrHo$_2$O$_4$, the zero-field ground state is similar to that of SrEr$_2$O$_4$, however, for the Ho1 sites, the magnetic order remains limited even at the lowest experimentally achievable temperature [39]. This lack of ordering can potentially be explained by the degeneracy (disorder) of both Ho1 and Ho2 subsystems in phase 13, however, it is also possible that for a non-Kramers Ho sites the crystal field effects lead to a considerable splitting of the ground state doublets at the lowest temperature.

In SrDy$_2$O$_4$, there are no long-range correlations between the magnetic moments in zero field, but they can be induced by applying a relatively weak magnetic field along the $b$ axis [12]. In fact the magnetization process in all the three compounds demonstrate similar features, as revealed by the low-$T$ single-crystal magnetization $M(H)$ measurements [7]. For certain directions of an applied field, the process is characterized by the appearance of a magnetization plateau, albeit not very pronounced but still clearly visible on the $dM(H)/dH$ curves. To stabilize the plateaus, the field should be applied along the $a$ axis in SrEr$_2$O$_4$ and along the $b$ axis in SrDy$_2$O$_4$ and SrHo$_2$O$_4$. The value of magnetization on the plateaux is approximately a third of the magnetization observed in higher fields [7]. The 1/3 magnetization plateaux are, of course, a common feature of many triangular antiferromagnets, they correspond to the states

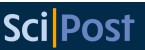

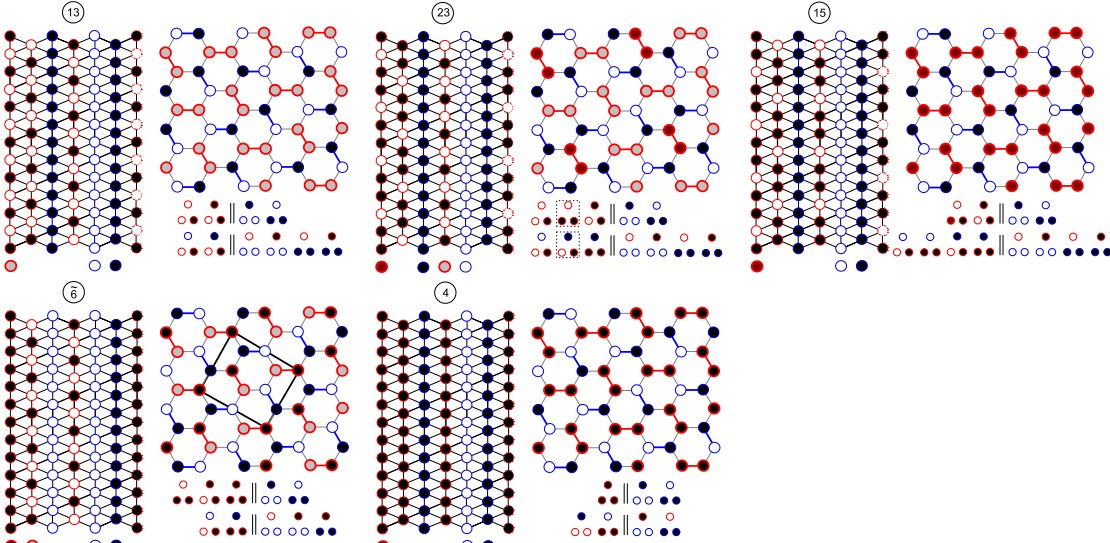

Figure 15: Sequence of the proposed phase transitions in SrHo$_2$O$_4$ and SrEr$_2$O$_4$ for a magnetic field applied along the easy-magnetization direction for the RE2 site, $a$ axis in SrEr$_2$O$_4$ and $b$ axis in SrHo$_2$O$_4$, for $J_2 < 0$, $J_{11} > 0$. For $J_2 > 0$ ($J_{11} > 0$), the phase $\widetilde{23}$ should appear instead of the phase 23 and the phase 6 should appear instead of the phase $\tilde{6}$. The corresponding sequence of magnetization values for the red sublattice is 0, 1/12, 1/3, 1/2, and 1 (per one site of red sublattice). Transition field values are $h_{1,13-23} = 2J_{01} - J_{11} + 4J_2$, $h_{1,23-15} = 2J_{01} - J_{11} - \frac{4}{3}J_2$, $h_{1,15-\tilde{6}} = 2J_{01} + 2J_{11} + 12J_2$, $h_{1,\tilde{6}-4} = 2J_{01} + 2J_{11} - 4J_2$. The width of the region $\tilde{6}$ is three times the width of the region 23 in ($h_1, h_2$)-plane (see Fig. 14).

with the two spins on each triangle pointing along the field and the third spin pointing in the opposite direction (the so called up-up-down, *uud*, structure) [40, 41]. Overall magnetization data are consistent with the Ising behavior in these three SrRE$_2$O$_4$ compounds. The two magnetic sites have their magnetization easy-axes aligned along (or very near) the two crystallographic axes, while when the field is applied along the third crystallographic axis, the measured magnetization is significantly lower (particularly for Ho and Er compounds [7]) suggesting that it is a hard magnetization axis for both sites.

Apart from the magnetization data, the evidence for a field-induced *uud* structure comes from the results of neutron diffraction for SrDy$_2$O$_4$ [12, 13], SrHo$_2$O$_4$ [14], SrEr$_2$O$_4$ [42] and BaDy$_2$O$_4$ [15]. The *uud* structures are characterized by the appearance of the sharp, almost resolution-limited magnetic peaks at non-integer positions. In SrHo$_2$O$_4$, the observed peaks are at the ($h0\frac{1}{3}$), ($h0\frac{2}{3}$) and symmetry related positions [14], in SrDy$_2$O$_4$, they are indexed by the propagation vector $\mathbf{k}' = [0 \ \frac{1}{3} \ \frac{1}{3}]$ [12, 13] and in BaDy$_2$O$_4$, the propagation vector is $\mathbf{k}' = [0 \ 0 \ \frac{1}{3}]$ [15].

Let us consider the case of a field induced 1/3 magnetization plateau – the field applied along the easy-magnetization direction for the RE2 site, $a$ axis in SrEr$_2$O$_4$ and $b$ axis in SrHo$_2$O$_4$ (see Fig. 15). The high-field phase with all the spins on the RE2 sites polarized along the field direction is phase 4. The experimentally determined *uud* structure is phase 15 (see Table 2 and Fig. 3). However, it follows from our study that regions 13 and 15 as well as regions 4 and 15 have no common 6-face. Therefore some intermediate phases should exist between them. These are probably phases 23 and $\tilde{6}$ ($\widetilde{23}$ and 6 if $J_2 > 0$) (see Fig. 15). Indeed the latest low-temperature magnetization and neutron diffraction measurements [42] indicated the presence of the additional intermediate phase(s) between zero-field and 1/3 magnetization

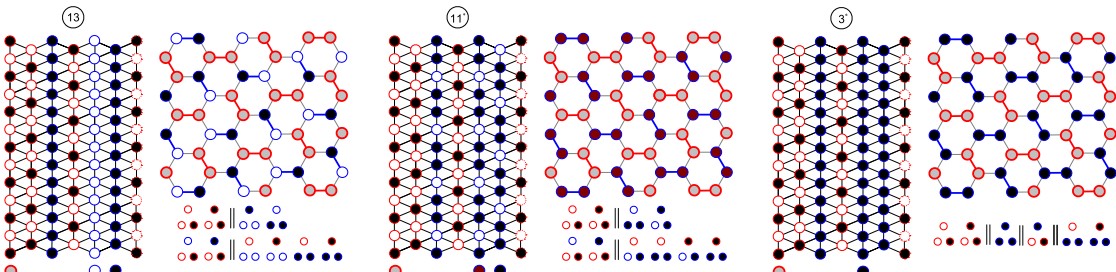

Figure 16: A sequence of phase transitions proposed for an increasing field applied along the $c$ axis in SrHo$_2$O$_4$ and SrEr$_2$O$_4$. The corresponding numbers for magnetization of the blue sublattice are 0, 1/3, and 1. $J_{02} > 0$. If $J_{02} < 0$, then there is a direct transition from phase 13 to phase 3$^*$.

plateau structures in SrEr$_2$O$_4$.

For $H \parallel b$ in SrDy$_2$O$_4$, the situation should be somewhat similar, but the low-field transition is from a disordered state and therefore difficult to describe within the framework of our theory. The transition from the field-induced *uud* structure into a fully polarized state should, however also involve an intermediate phase. For $H \parallel b$ in SrDy$_2$O$_4$, the zero-field phase is the disordered phase 14. There are several possibilities for the field-induced *uud* phase from phase 14 (see Appendix B and Fig. 14). The transition from the *uud* structure into a fully polarized state should also involve an intermediate phase.

A very interesting case is found in BaDy$_2$O$_4$. Its low temperature ($T < T_N = 0.48$ K) zero-field structure is characterized by two half-integer propagation vectors, $\mathbf{k}_1 = [\frac{1}{2}\ 0\ \frac{1}{2}]$ and $\mathbf{k}_2 = [\frac{1}{2}\ \frac{1}{2}\ \frac{1}{2}]$ [6], suggesting stabilization of a significantly different ground state compared to SrEr$_2$O$_4$, SrHo$_2$O$_4$, and SrDy$_2$O$_4$. An in-field behaviour, however, seems to be rather similar, as a *uud* structure is again inferred from powder neutron diffraction measurements and from a pronounced plateau in the magnetization curve [15]. The *uud* structure appears to be much more stable than the zero field states, as it survives warming to the three times higher temperature than the $T_N(H = 0)$. Remarkably, the field-induced magnetic structure depicted in Fig. 7 of Ref. [15] contains the triangles with all 3 magnetic moments pointing in the same direction (the ladders with *uuuudd* structure). This structure cannot be energetically favorable if only non-interacting zigzag ladders are considered, but present in our model (structures 7 or 10). Unfortunately, finding any further intermediate magnetic states in BaDy$_2$O$_4$ is experimentally challenging in the absence of large single crystal samples of this compound.

Let us conclude this section by considering the case of a field applied along the $c$ axis (direction of the chains of the magnetic atoms). For this geometry, magnetization data for SrHo$_2$O$_4$ and SrEr$_2$O$_4$ suggest a single phase transition to a state with a full polarization of a site for which the easy-magnetization direction coincides with the $c$ axis. In the language of this paper, the transition is from a zero-field phase 13 to phase 3$^*$ where all spins on one of the magnetic sites are parallel to the field while the other site remains the same as in zero-field. The proposal is that with increasing field, structure 11$^*$ is stabilized between phases 13 and 3$^*$ (see Fig. 16), although a direct transition between these phases is also possible.

## 4   Conclusions

We present a solution to the ground-state problem for an Ising model in an external field for a honeycomb zigzag-ladder lattice with two different types of magnetic sites. Although the solution is incomplete, the presence of a variety of ground-states is proved and, for several phases,

the corresponding regions in seven-dimensional parameter space are completely determined. Some of these phases are ordered but the majority are disordered with the disorder being one, two, or even three-dimensional.

The solution is used to explain the formation of experimentally determined spin arrangements in honeycomb zigzag-ladder magnets $SrRE_2O_4$ and $BaRE_2O_4$ in an applied magnetic field. In particularly for $SrEr_2O_4$ and $BaDy_2O_4$, the solution predicts new magnetic phases (with two different types of magnetic configurations on the same ladder), recently found or yet to be detected experimentally.

Since the set of basic rays that we found here is incomplete, we hope that the paper will inspire further efforts to find the remaining basic rays and to establish a complete set for this very interesting and complex ground-state problem.

## Acknowledgments

Yu. D. acknowledges the hospitality by the Max Planck Institute for the Physics of Complex Systems in Dresden and a support from the University of Warwick Materials Global Research Priority.

## A   Appendix: Energy of triangular plaquette configurations

Here we present the energies for all the six configurations, $\overset{\circ}{\circ\circ}$, $\overset{\circ}{\circ\bullet}$, $\overset{\bullet}{\circ\circ}$, $\overset{\circ}{\bullet\bullet}$, $\overset{\bullet}{\circ\bullet}$, and $\overset{\bullet}{\bullet\bullet}$ (open and solid circles denote spins $\sigma = -1$ and $\sigma = +1$, respectively), of the four types of plaquettes (see Fig. 2 and Eq. (2)).

$$
\begin{aligned}
e_{11} &= (1-\alpha_1)J_{01} + J_{11} + [(1-\eta_1)\gamma_1 + (1-\delta_1)(1-\gamma_1)]h_1, \\
e_{12} &= -(1-\alpha_1)J_{01} + (1-\delta_1)(1-\gamma_1)h_1, \\
e_{13} &= (1-\alpha_1)J_{01} - J_{11} + [(1-\eta_1)\gamma_1 - (1-\delta_1)(1-\gamma_1)]h_1, \\
e_{14} &= (1-\alpha_1)J_{01} - J_{11} - [(1-\eta_1)\gamma_1 - (1-\delta_1)(1-\gamma_1)]h_1, \\
e_{15} &= -(1-\alpha_1)J_{01} - (1-\delta_1)(1-\gamma_1)h_1, \\
e_{16} &= (1-\alpha_1)J_{01} + J_{11} - [(1-\eta_1)\gamma_1 + (1-\delta_1)(1-\gamma_1)]h_1;
\end{aligned}
\tag{A.1}
$$

$$
\begin{aligned}
e_{21} &= (1-\alpha_2)J_{02} + J_{12} + [(1-\eta_2)\gamma_2 + (1-\delta_2)(1-\gamma_2)]h_2, \\
e_{22} &= -(1-\alpha_2)J_{02} + (1-\delta_2)(1-\gamma_2)h_2, \\
e_{23} &= (1-\alpha_2)J_{02} - J_{12} + [(1-\eta_2)\gamma_2 - (1-\delta_2)(1-\gamma_2)]h_2, \\
e_{24} &= (1-\alpha_2)J_{02} - J_{12} - [(1-\eta_2)\gamma_2 - (1-\delta_2)(1-\gamma_2)]h_2, \\
e_{25} &= -(1-\alpha_2)J_{02} - (1-\delta_2)(1-\gamma_2)h_2, \\
e_{26} &= (1-\alpha_2)J_{02} + J_{12} - [(1-\eta_2)\gamma_2 + (1-\delta_2)(1-\gamma_2)]h_2;
\end{aligned}
\tag{A.2}
$$

$$e_{31} = \frac{\alpha_1}{2}J_{01} + 2(1-\beta)J_2 + \eta_1\frac{\gamma_1}{2}h_1 + \delta_2\frac{(1-\gamma_2)}{2}h_2,$$

$$e_{32} = -\frac{\alpha_1}{2}J_{01} + \delta_2\frac{(1-\gamma_2)}{2}h_2,$$

$$e_{33} = \frac{\alpha_1}{2}J_{01} - 2(1-\beta)J_2 + \eta_1\frac{\gamma_1}{2}h_1 - \delta_2\frac{(1-\gamma_2)}{2}h_2,$$

$$e_{34} = \frac{\alpha_1}{2}J_{01} - 2(1-\beta)J_2 - \eta_1\frac{\gamma_1}{2}h_1 + \delta_2\frac{(1-\gamma_2)}{2}h_2,$$

$$e_{35} = -\frac{\alpha_1}{2}J_{01} - \delta_2\frac{(1-\gamma_2)}{2}h_2,$$

$$e_{36} = \frac{\alpha_1}{2}J_{01} + 2(1-\beta)J_2 - \eta_1\frac{\gamma_1}{2}h_1 - \delta_2\frac{(1-\gamma_2)}{2}h_2; \tag{A.3}$$

$$e_{41} = \frac{\alpha_2}{2}J_{02} + 2\beta J_2 + \delta_1\frac{(1-\gamma_1)}{2}h_1 + \eta_2\frac{\gamma_2}{2}h_2,$$

$$e_{42} = -\frac{\alpha_2}{2}J_{02} + \delta_1\frac{(1-\gamma_1)}{2}h_1,$$

$$e_{43} = \frac{\alpha_2}{2}J_{02} - 2\beta J_2 - \delta_1\frac{(1-\gamma_1)}{2}h_1 + \eta_2\frac{\gamma_2}{2}h_2,$$

$$e_{44} = \frac{\alpha_2}{2}J_{02} - 2\beta J_2 + \delta_1\frac{(1-\gamma_1)}{2}h_1 - \eta_2\frac{\gamma_2}{2}h_2,$$

$$e_{45} = -\frac{\alpha_2}{2}J_{02} - \delta_1\frac{(1-\gamma_1)}{2}h_1,$$

$$e_{46} = \frac{\alpha_2}{2}J_{02} + 2\beta J_2 - \delta_1\frac{(1-\gamma_1)}{2}h_1 - \eta_2\frac{\gamma_2}{2}h_2. \tag{A.4}$$

To calculate the energy of a structure (or structures in the case of degeneracy), it is sufficient to know the relative numbers of plaquette configurations which generate this structure. For instance, structures 2 are generated with seven configurations 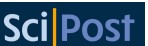, and (see Table 2), relative numbers of which in these structures are 3, 1, 2, 2, 4, 4, and 2, respectively. Hence, the energy (per six plaquettes) of structures 2 is

$$e_2 = \frac{1}{3}(3e_{16} + e_{24} + 2e_{25} + 2e_{34} + 4e_{36} + 4e_{45} + 2e_{46})$$

$$= \frac{1}{3}(3J_{01} - J_{02} + 3J_{11} - J_{12} + 4J_2 - 3h_1 - h_2). \tag{A.5}$$

It should be noted that this energy does not depend on free coefficients although $e_{ij}$ do depend on these. The magnetization of the red sublattice (per one red site) is equal to $3 \cdot 1/3 = 1$, for the blue sublattice, it is equal to $1/3$.

Let us show how to find conditions for the existence of a region in the $(h_1, h_2)$-plane, for instance, region 4. This region is determined with the set of basic rays $\{\mathbf{r}_1, \mathbf{r}_1^\star, \mathbf{r}_2, \mathbf{r}_3^\star, \mathbf{r}_5, \mathbf{r}_6, \mathbf{r}_6^\star, \mathbf{r}_6^{\star-}, \mathbf{r}_7, \mathbf{r}_7^{\sim}\}$.

From the equation

$$a_1\mathbf{r}_1 + a_1^\star\mathbf{r}_1^\star + a_2\mathbf{r}_2 + a_3^\star\mathbf{r}_3^\star + a_5\mathbf{r}_5 + a_6\mathbf{r}_6 + a_6^\star\mathbf{r}_6^\star + a_6^{\star-}\mathbf{r}_6^{\star-} + a_7\mathbf{r}_7 + a_7^{\sim}\mathbf{r}_7^{\sim}$$
$$= (J_{01}, J_{02}, J_{11}, J_{12}, J_2, h_1, h_2), \tag{A.6}$$

we have

$$a_1 = a_2 + a_5 - J_{01}, \quad a_1^\star = a_3^\star - J_{02},$$
$$a_6 = 2a_2 + J_{11}, \quad a_6^{\star-} = -2a_3^\star - a_6^\star + J_{12}, \quad a_7 = a_7^{\sim} + J_2,$$
$$h_1 = 4a_2 + 2a_5 + 8a_7^{\sim} + 2J_{11} + 4J_2, \quad h_2 = 4a_3^\star + 4a_6^\star - 2J_{12} + 4J_2. \tag{A.7}$$

All the coefficients $a$ should be nonnegative, therefore, we obtain

$$J_{12} > 0, \quad -2J_{02} + J_{12} > 0. \tag{A.8}$$

These inequalities are the conditions for the existence of region 4 in $h_1, h_2$-plain at fixed $J_{01}$, $J_{02}, J_{11} J_{12}$, and $J_2$. For $h_1$ and $h_2$ of region 4 we have,

$$
\begin{aligned}
h_1 &> 2J_{11} + 4|J_2| & &\text{if } J_{01} < 0, \\
h_1 &> 2J_{01} + J_{11} + 4|J_2| & &\text{if } J_{01} > 0,\ J_{11} < 0, \\
h_1 &> 2J_{01} + 2J_{11} + 4|J_2| & &\text{if } J_{01} > 0,\ J_{11} > 0, \\
-2J_{12} + 4J_2 &< h_2 < 2J_{12} + 4J_2 & &\text{if } J_{02} < 0, \\
4J_{02} - 2J_{12} + 4J_2 &< h_2 < -4J_{02} + 2J_{12} & &\text{if } J_{02} > 0.
\end{aligned}
\tag{A.9}
$$

## B  Appendix: Completeness of sets of basic rays

Considering a set of basic rays (vectors) for a fully dimensional phase (see Table 2), we can check whether this set is complete. All linear combinations with nonnegative coefficients of the basic rays form a seven-dimensional polyhedral cone (see Subsection 2.1). First, we should find all the six-dimensional faces (6-faces) of the polyhedral cone and the configurations of triangular plaquettes for these 6-faces. As an example, let us consider phase 13. The set of basic vectors for this phase is $\{\mathbf{r}_1^\star, \mathbf{r}_2, \mathbf{r}_3, \mathbf{r}_3^\star, \mathbf{r}_4, \widetilde{\mathbf{r}_4}, \mathbf{r}_5, \mathbf{r}_5^-, \mathbf{r}_6^\star, \mathbf{r}_6^{\star-}\}$. The sets of basic vectors for its 6-faces (enumerated from 1 to 12) and corresponding sets of plaquette configurations are given below. For each 6-face the corresponding neighboring phase is indicated in parentheses on the right.

(1) $\{\mathbf{r}_1^\star, \mathbf{r}_2, \mathbf{r}_3, \mathbf{r}_4, \widetilde{\mathbf{r}_4}, \mathbf{r}_5, \mathbf{r}_5^-, \mathbf{r}_6^\star\}$,  $(3^\star, 13)$

(2) $\{\mathbf{r}_1^\star, \mathbf{r}_2, \mathbf{r}_3, \mathbf{r}_4, \widetilde{\mathbf{r}_4}, \mathbf{r}_5, \mathbf{r}_5^-, \mathbf{r}_6^{\star-}\}$,  $(\overline{3}^\star, 13)$

(3) $\{\mathbf{r}_2, \mathbf{r}_3, \mathbf{r}_3^\star, \mathbf{r}_4, \widetilde{\mathbf{r}_4}, \mathbf{r}_5, \mathbf{r}_5^-, \mathbf{r}_6^\star\}$,  $(11^\star, 13)$

(4) $\{\mathbf{r}_2, \mathbf{r}_3, \mathbf{r}_3^\star, \mathbf{r}_4, \widetilde{\mathbf{r}_4}, \mathbf{r}_5, \mathbf{r}_5^-, \mathbf{r}_6^{\star-}\}$,  $(\overline{11}^\star, 13)$

(5) $\{\mathbf{r}_1^\star, \mathbf{r}_2, \mathbf{r}_3^\star, \widetilde{\mathbf{r}_4}, \mathbf{r}_5, \mathbf{r}_6^\star, \mathbf{r}_6^{\star-}\}$,  $(22, 13)$

(6) $\{\mathbf{r}_1^\star, \mathbf{r}_2, \mathbf{r}_3^\star, \widetilde{\mathbf{r}_4}, \mathbf{r}_5^-, \mathbf{r}_6^\star, \mathbf{r}_6^{\star-}\}$,  $(\overline{22}, 13)$

(7) $\{\mathbf{r}_1^\star, \mathbf{r}_2, \mathbf{r}_3^\star, \mathbf{r}_4, \mathbf{r}_5, \mathbf{r}_6^\star, \mathbf{r}_6^{\star-}\}$,  $(\widetilde{22}, 13)$

(8) $\{\mathbf{r}_1^\star, \mathbf{r}_2, \mathbf{r}_3^\star, \mathbf{r}_4, \mathbf{r}_5^-, \mathbf{r}_6^\star, \mathbf{r}_6^{\star-}\}$,  $(\overline{\widetilde{22}}, 13)$

(9) $\{\mathbf{r}_1^\star, \mathbf{r}_3, \mathbf{r}_3^\star, \widetilde{\mathbf{r}_4}, \mathbf{r}_5, \mathbf{r}_6^\star, \mathbf{r}_6^{\star-}\}$,  $(23, 13)$

(10) $\{\mathbf{r}_1^\star, \mathbf{r}_3, \mathbf{r}_3^\star, \widetilde{\mathbf{r}_4}, \mathbf{r}_5^-, \mathbf{r}_6^\star, \mathbf{r}_6^{\star-}\}$,  $(\overline{23}, 13)$

(11) $\{\mathbf{r}_1^\star, \mathbf{r}_3, \mathbf{r}_3^\star, \mathbf{r}_4, \mathbf{r}_5, \mathbf{r}_6^\star, \mathbf{r}_6^{\star-}\}$,  $(\widetilde{23}, 13)$

(12) $\{\mathbf{r}_1^\star, \mathbf{r}_3, \mathbf{r}_3^\star, \mathbf{r}_4, \mathbf{r}_5^-, \mathbf{r}_6^\star, \mathbf{r}_6^{\star-}\}$.  $(\overline{\widetilde{23}}, 13)$

In the 6-faces (1) to (4), in addition to the structures 13, there are other structures that we have previously identified. This means that these 6-faces indeed bound the region for phase 13. The remaining eight 6-faces, (5) to (12), also bound this region, since, in addition to structures 13, some new structures can be constructed for these 6-faces. The structures (among these new ones) containing the greatest number of plaquette configurations which are

absent in structures 13 [configuration ⦿ and ⦿ for the 6-face (5); configurations ⦿ and ⦿ for the 6-face (9)] are the structures of fully dimensional phases whose regions have common boundaries with region 13. These structures are shown in Fig. 9 and we have therefore found the complete set of basic vectors for phase 13.

The complete sets of basic vectors are also found for phases 4 ($\{\mathbf{r}_1, \mathbf{r}_1^\star, \mathbf{r}_2, \mathbf{r}_3^\star, \mathbf{r}_5, \mathbf{r}_6, \mathbf{r}_6^\star, \mathbf{r}_6^{\star-}, \mathbf{r}_7, \widetilde{\mathbf{r}_7}\}$), 11 ($\{\mathbf{r}_2^\star, \mathbf{r}_3, \mathbf{r}_3^\star, \mathbf{r}_4^\star, \widetilde{\mathbf{r}_4}^\star, \mathbf{r}_5, \mathbf{r}_5^\star, \mathbf{r}_5^{\star-}, \mathbf{r}_6\}$), and 14 ($\{\mathbf{r}_2, \mathbf{r}_2^\star, \mathbf{r}_3, \mathbf{r}_3^\star, \mathbf{r}_4, \widetilde{\mathbf{r}_4}, \mathbf{r}_4^\star, \widetilde{\mathbf{r}_4}^\star, \mathbf{r}_5, \mathbf{r}_5^-, \mathbf{r}_5^\star, \mathbf{r}_5^{\star-}\}$). For the rest of the phases the sets of basic vectors are incomplete. Consider, for instance, phase 10. Its set of basic vectors, $\{\mathbf{r}_2^\star, \mathbf{r}_3, \widetilde{\mathbf{r}_4}^\star, \widetilde{\mathbf{r}_4}^{\star}, \mathbf{r}_5, \mathbf{r}_5^\star, \mathbf{r}_6\}$, is incomplete. The sets of basic vectors for the 6-faces of the corresponding polyhedral cone and sets of plaquette configurations are as follows

(1) $\{\mathbf{r}_2^\star, \widetilde{\mathbf{r}_4}, \widetilde{\mathbf{r}_4}^{\star}, \mathbf{r}_5, \mathbf{r}_5^\star, \mathbf{r}_6\}$, $(1,10)$

(2) $\{\mathbf{r}_2^\star, \mathbf{r}_3, \widetilde{\mathbf{r}_4}, \mathbf{r}_5, \mathbf{r}_5^\star, \mathbf{r}_6\}$, $(2^\star,10)$

(3) $\{\mathbf{r}_3, \widetilde{\mathbf{r}_4}, \widetilde{\mathbf{r}_4}^{\star}, \mathbf{r}_5, \mathbf{r}_5^\star, \mathbf{r}_6\}$, $(9,10)$

(4) $\{\mathbf{r}_2^\star, \mathbf{r}_3, \widetilde{\mathbf{r}_4}^{\star}, \mathbf{r}_5, \mathbf{r}_5^\star, \mathbf{r}_6\}$, $(11,10)$

(5) $\{\mathbf{r}_2^\star, \mathbf{r}_3, \widetilde{\mathbf{r}_4}, \widetilde{\mathbf{r}_4}^{\star}, \mathbf{r}_5, \mathbf{r}_5^\star\}$, $(14,10)$

(6) $\{\mathbf{r}_2^\star, \mathbf{r}_3, \widetilde{\mathbf{r}_4}, \widetilde{\mathbf{r}_4}^{\star}, \mathbf{r}_5, \mathbf{r}_6\}$, $(20,10)$

(7) $\{\mathbf{r}_2^\star, \mathbf{r}_3, \widetilde{\mathbf{r}_4}, \widetilde{\mathbf{r}_4}^{\star}, \mathbf{r}_5, \mathbf{r}_6\}$, $(-,10)$

6-face (6) gives a new phase, "nontriangular" phase 20 (Fig. 9). 6-face (7) of the polyhedral cone is not a 6-face between two fully dimensional phase regions because, with the corresponding set of triangular configurations, it is not possible to construct any structure different from structures 10 (although it is possible to construct a new configuration of a "hexagonal tube"). Therefore the set of basic vectors for phase 10 is incomplete.

Here we give the complete sets of basic vectors for phases 4, 11, and 14, their 6-faces and corresponding sets of plaquette configurations. For each 6-face the neighboring phase is indicated in parentheses on the right.

Phase 4    $\{\mathbf{r}_1, \mathbf{r}_1^\star, \mathbf{r}_2, \mathbf{r}_3^\star, \mathbf{r}_5, \mathbf{r}_6, \mathbf{r}_6^\star, \mathbf{r}_6^{\star-}, \mathbf{r}_7, \widetilde{\mathbf{r}_7}\}$

$\{\mathbf{r}_1, \mathbf{r}_1^\star, \mathbf{r}_2, \mathbf{r}_5, \mathbf{r}_6, \mathbf{r}_6^\star, \mathbf{r}_7, \widetilde{\mathbf{r}_7}\}$, $(1,4)$

$\{\mathbf{r}_1, \mathbf{r}_1^\star, \mathbf{r}_2, \mathbf{r}_5, \mathbf{r}_6, \mathbf{r}_6^{\star-}, \mathbf{r}_7, \widetilde{\mathbf{r}_7}\}$, $(\widetilde{1},4)$

$\{\mathbf{r}_1, \mathbf{r}_2, \mathbf{r}_3^\star, \mathbf{r}_5, \mathbf{r}_6, \mathbf{r}_6^\star, \mathbf{r}_7, \widetilde{\mathbf{r}_7}\}$, $(2,4)$

$\{\mathbf{r}_1, \mathbf{r}_2, \mathbf{r}_3^\star, \mathbf{r}_5, \mathbf{r}_6, \mathbf{r}_6^{\star-}, \mathbf{r}_7, \widetilde{\mathbf{r}_7}\}$, $(\widetilde{2},4)$

$\{\mathbf{r}_1^\star, \mathbf{r}_3^\star, \mathbf{r}_5, \mathbf{r}_6, \mathbf{r}_6^\star, \mathbf{r}_6^{\star-}, \mathbf{r}_7\}$, $(6,4)$

$\{\mathbf{r}_1^\star, \mathbf{r}_3^\star, \mathbf{r}_5, \mathbf{r}_6, \mathbf{r}_6^\star, \mathbf{r}_6^{\star-}, \widetilde{\mathbf{r}_7}\}$, $(\widetilde{6},4)$

$\{\mathbf{r}_1, \mathbf{r}_1^\star, \mathbf{r}_3^\star, \mathbf{r}_6, \mathbf{r}_6^\star, \mathbf{r}_6^{\star-}, \mathbf{r}_7\}$, $(12,4)$

$\{\mathbf{r}_1, \mathbf{r}_1^\star, \mathbf{r}_3^\star, \mathbf{r}_6, \mathbf{r}_6^\star, \mathbf{r}_6^{\star-}, \widetilde{\mathbf{r}_7}\}$, $(\widetilde{12},4)$

$\{\mathbf{r}_1^\star, \mathbf{r}_2, \mathbf{r}_3^\star, \mathbf{r}_5, \mathbf{r}_6, \mathbf{r}_6^{\star-}, \mathbf{r}_7\}$, $(18,4)$

$\{\mathbf{r}_1^\star, \mathbf{r}_2, \mathbf{r}_3^\star, \mathbf{r}_5, \mathbf{r}_6^\star, \mathbf{r}_6^{\star-}, \widetilde{\mathbf{r}_7}\}$, $(\widetilde{18},4)$

$\{\mathbf{r}_1, \mathbf{r}_1^\star, \mathbf{r}_2, \mathbf{r}_3^\star, \mathbf{r}_6^\star, \mathbf{r}_6^{\star-}, \widetilde{\mathbf{r}_7}\}$, $(19,4)$

$\{\mathbf{r}_1, \mathbf{r}_1^\star, \mathbf{r}_2, \mathbf{r}_3^\star, \mathbf{r}_6^\star, \mathbf{r}_6^{\star-}, \mathbf{r}_7\}$. $(\widetilde{19},4)$

Phase 11    $\{\mathbf{r}_2^\star, \mathbf{r}_3, \mathbf{r}_3^\star, \mathbf{r}_4^\star, \widetilde{\mathbf{r}_4}^{\star}, \mathbf{r}_5, \mathbf{r}_5^\star, \mathbf{r}_5^{\star-}, \mathbf{r}_6\}$

$\{r_2^\star, r_3^\star, r_4^\star, r_4^{\widetilde{\star}}, r_5, r_5^\star, r_5^{\star-}, r_6\},\quad (3, 11)$

$\{r_3, r_3^\star, r_4^\star, r_5, r_5^\star, r_6\},\quad (8, 11)$

$\{r_3, r_3^\star, r_4^{\widetilde{\star}}, r_5, r_5^{\star-}, r_6\},\quad (\bar{8}, 11)$

$\{r_3, r_3^\star, r_4^{\widetilde{\star}}, r_5, r_5^\star, r_6\},\quad (9, 11)$

$\{r_3, r_3^\star, r_4^\star, r_5, r_5^{\star-}, r_6\},\quad (\widetilde{9}, 11)$

$\{r_2^\star, r_3, r_4^{\widetilde{\star}}, r_5, r_5^\star, r_6\},\quad (10, 11)$

$\{r_2^\star, r_3, r_4^\star, r_5, r_5^{\star-}, r_6\},\quad (\widetilde{10}, 11)$

$\{r_2^\star, r_3, r_3^\star, r_4^\star, r_4^{\widetilde{\star}}, r_5^\star, r_5^{\star-}, r_6\},\quad (13^\star, 11)$

$\{r_2^\star, r_3, r_3^\star, r_4^\star, r_4^{\widetilde{\star}}, r_5, r_5^\star, r_5^{\star-}\},\quad (14, 11)$

$\{r_2^\star, r_3, r_4^\star, r_5, r_5^\star, r_6\},\quad (21, 11)$

$\{r_2^\star, r_3, r_4^{\widetilde{\star}}, r_5, r_5^{\star-}, r_6\}.\quad (\widetilde{21}, 11)$

Phase 14  $\{r_2, r_2^\star, r_3, r_3^\star, r_4, r_4^{\widetilde{}}, r_4^\star, r_4^{\widetilde{\star}}, r_5, r_5^-, r_5^\star, r_5^{\star-}\}$

$\{r_2, r_2^\star, r_4^{\widetilde{}}, r_4^{\widetilde{\star}}, r_5, r_5^\star\},\quad (1, 14)$

$\{r_2, r_2^\star, r_4^{\widetilde{}}, r_4^{\widetilde{\star}}, r_5^-, r_5^{\star-}\},\quad (\bar{1}, 14)$

$\{r_2, r_2^\star, r_4, r_4^\star, r_5, r_5^{\star-}\},\quad (\widetilde{1}, 14)$

$\{r_2, r_2^\star, r_4, r_4^\star, r_5^-, r_5^\star\},\quad (\bar{\widetilde{1}}, 14)$

$\{r_2, r_2^\star, r_3^\star, r_4^\star, r_4^{\widetilde{\star}}, r_5, r_5^\star, r_5^{\star-}\},\quad (3, 14)$

$\{r_2, r_2^\star, r_3^\star, r_4^\star, r_4^{\widetilde{\star}}, r_5^-, r_5^\star, r_5^{\star-}\},\quad (\bar{3}, 14)$

$\{r_2, r_2^\star, r_3, r_4, r_4^{\widetilde{}}, r_5, r_5^-, r_5^\star\},\quad (3^\star, 14)$

$\{r_2, r_2^\star, r_3, r_4, r_4^{\widetilde{}}, r_5, r_5^-, r_5^{\star-}\},\quad (\bar{3}^\star, 14)$

$\{r_2, r_2^\star, r_4, r_4^\star, r_5, r_5^\star\},\quad (7, 14)$

$\{r_2, r_2^\star, r_4, r_4^\star, r_5^-, r_5^{\star-}\},\quad (\bar{7}, 14)$

$\{r_2, r_2^\star, r_4^{\widetilde{}}, r_4^{\widetilde{\star}}, r_5, r_5^{\star-}\},\quad (\widetilde{7}, 14)$

$\{r_2, r_2^\star, r_4^{\widetilde{}}, r_4^{\widetilde{\star}}, r_5^-, r_5^\star\},\quad (\bar{\widetilde{7}}, 14)$

$\{r_3, r_3^\star, r_4, r_4^\star, r_5, r_5^\star\},\quad (8, 14)$

$\{r_3, r_3^\star, r_4, r_4^\star, r_5^-, r_5^{\star-}\},\quad (\bar{8}, 14)$

$\{r_3, r_3^\star, r_4^{\widetilde{}}, r_4^{\widetilde{\star}}, r_5, r_5^{\star-}\},\quad (\widetilde{8}, 14)$

$\{r_3, r_3^\star, r_4^{\widetilde{}}, r_4^{\widetilde{\star}}, r_5^-, r_5^\star\},\quad (\bar{\widetilde{8}}, 14)$

$\{r_3, r_3^\star, r_4^{\widetilde{}}, r_4^{\widetilde{\star}}, r_5, r_5^\star\},\quad (9, 14)$

$\{r_3, r_3^\star, r_4^{\widetilde{}}, r_4^{\widetilde{\star}}, r_5^-, r_5^{\star-}\},\quad (\bar{9}, 14)$

$\{r_3, r_3^\star, r_4, r_4^\star, r_5, r_5^{\star-}\},\quad (\widetilde{9}, 14)$

$\{r_3, r_3^\star, r_4, r_4^\star, r_5^-, r_5^\star\},\quad (\bar{\widetilde{9}}, 14)$

$\{r_2^\star, r_3, r_4^{\widetilde{}}, r_4^{\widetilde{\star}}, r_5, r_5^\star\},\quad (10, 14)$

$\{r_2^\star, r_3, r_4^{\widetilde{}}, r_4^{\widetilde{\star}}, r_5^-, r_5^{\star-}\},\quad (\widetilde{10}, 14)$

$\{\mathbf{r}_2, \mathbf{r}_3^\star, \mathbf{r}_4^\sim, \mathbf{r}_4^{\sim\star}, \mathbf{r}_5, \mathbf{r}_5^\star\}$, $(10^\star, 14)$

$\{\mathbf{r}_2, \mathbf{r}_3^\star, \mathbf{r}_4^\sim, \mathbf{r}_4^{\sim\star}, \mathbf{r}_5^-, \mathbf{r}_5^{\star-}\}$, $(\overline{10^\star}, 14)$

$\{\mathbf{r}_2^\star, \mathbf{r}_3, \mathbf{r}_4, \mathbf{r}_4^\star, \mathbf{r}_5, \mathbf{r}_5^{\star-}\}$, $(\widetilde{10}, 14)$

$\{\mathbf{r}_2^\star, \mathbf{r}_3, \mathbf{r}_4, \mathbf{r}_4^\star, \mathbf{r}_5^-, \mathbf{r}_5^\star\}$, $(\overline{\widetilde{10}}, 14)$

$\{\mathbf{r}_2, \mathbf{r}_3^\star, \mathbf{r}_4, \mathbf{r}_4^\star, \mathbf{r}_5, \mathbf{r}_5^{\star-}\}$, $(^\star\widetilde{10}, 14)$

$\{\mathbf{r}_2, \mathbf{r}_3^\star, \mathbf{r}_4, \mathbf{r}_4^\star, \mathbf{r}_5^-, \mathbf{r}_5^\star\}$, $(^\star\overline{\widetilde{10}}, 14)$

$\{\mathbf{r}_2^\star, \mathbf{r}_3, \mathbf{r}_3^\star, \mathbf{r}_4^\star, \mathbf{r}_4^{\sim\star}, \mathbf{r}_5, \mathbf{r}_5^\star, \mathbf{r}_5^{\star-}\}$, $(11, 14)$

$\{\mathbf{r}_2^\star, \mathbf{r}_3, \mathbf{r}_3^\star, \mathbf{r}_4^\star, \mathbf{r}_4^{\sim\star}, \mathbf{r}_5^-, \mathbf{r}_5^\star, \mathbf{r}_5^{\star-}\}$, $(\overline{11}, 14)$

$\{\mathbf{r}_2, \mathbf{r}_3, \mathbf{r}_3^\star, \mathbf{r}_4, \mathbf{r}_4^\sim, \mathbf{r}_5, \mathbf{r}_5^-, \mathbf{r}_5^\star\}$, $(11^\star, 14)$

$\{\mathbf{r}_2, \mathbf{r}_3, \mathbf{r}_3^\star, \mathbf{r}_4, \mathbf{r}_4^\sim, \mathbf{r}_5, \mathbf{r}_5^-, \mathbf{r}_5^\star\}$, $(\overline{11}^\star, 14)$

$\{\mathbf{r}_2^\star, \mathbf{r}_3, \mathbf{r}_4, \mathbf{r}_4^\star, \mathbf{r}_5, \mathbf{r}_5^\star\}$, $(24, 14)$

$\{\mathbf{r}_2^\star, \mathbf{r}_3, \mathbf{r}_4, \mathbf{r}_4^\star, \mathbf{r}_5^-, \mathbf{r}_5^\star\}$, $(\overline{24}, 14)$

$\{\mathbf{r}_2, \mathbf{r}_3^\star, \mathbf{r}_4, \mathbf{r}_4^\star, \mathbf{r}_5, \mathbf{r}_5^\star\}$, $(24^\star, 14)$

$\{\mathbf{r}_2, \mathbf{r}_3^\star, \mathbf{r}_4, \mathbf{r}_4^\star, \mathbf{r}_5^-, \mathbf{r}_5^{\star-}\}$, $(\overline{24}^\star, 14)$

$\{\mathbf{r}_2^\star, \mathbf{r}_3, \mathbf{r}_4^\sim, \mathbf{r}_4^{\sim\star}, \mathbf{r}_5, \mathbf{r}_5^{\star-}\}$, $(\widetilde{24}, 14)$

$\{\mathbf{r}_2^\star, \mathbf{r}_3, \mathbf{r}_4^\sim, \mathbf{r}_4^{\sim\star}, \mathbf{r}_5^-, \mathbf{r}_5^\star\}$, $(\overline{\widetilde{24}}, 14)$

$\{\mathbf{r}_2, \mathbf{r}_3^\star, \mathbf{r}_4^\sim, \mathbf{r}_4^{\sim\star}, \mathbf{r}_5, \mathbf{r}_5^{\star-}\}$, $(^\star\widetilde{24}, 14)$

$\{\mathbf{r}_2, \mathbf{r}_3^\star, \mathbf{r}_4^\sim, \mathbf{r}_4^{\sim\star}, \mathbf{r}_5^-, \mathbf{r}_5^\star\}$. $(^\star\overline{\widetilde{24}}, 14)$

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
