# Peer review of "An Ising model on a 3D honeycomb zigzag-ladder lattice: a solution to the ground-state problem"

_SciPost Physics Core, doi:SciPost Phys. Core 5, 047 (2022)_

## Round 1 · Referee Report · Anonymous (Referee 1) · 2021-8-4

Report

Reviewer report on the manuscript scipost_202107_00007v1: "Ising model on a 3D honeycomb zigzag-ladder lattice: a solution to the ground-state problem and application to the SrRE2O4 and BaRE2O4 magnets" by Yu. I. Dublenych and O. A. Petrenko

The submitted paper brings an incomplete exact solution for a ground state of the spin-1/2 Ising model on a 3D honeycomb zigzag-ladder lattice, which is treated by a method of basic rays and sets of cluster configurations. The investigated model is inspired by rare-earth magnetic compounds SrRE2O4 and BaRE2O4, which are definitely worthwhile to study. However, the manuscript has following serious insufficiencies and drawbacks:

1. The solution for a ground-state problem of the spin-1/2 Ising model on a 3D honeycomb zigzag-ladder lattice is incomplete and hence, one cannot exclude possibility of overlooking some additional ground state(s) and overall incompleteness of the established ground-state phase diagrams.
2. The used method has been developed by one of the present authors more than 10 years ago and because of its obscurity it did not find any followers yet (at least to the best of my knowledge). From this perspective, it seems to be of marginal interest for potential readers from the methodological point of view.
3. The investigated model was developed for rare-earth magnets SrRE2O4 and BaRE2O4, however, the paper contains only a little new information about these compounds. The authors merely predict character of the ground states, but their stability (field range) is not comprehensively compared with available experimental data. The true nature of ground state is also questionable.
4. The appropriateness of the suggested model for rare-earth magnets SrRE2O4 and BaRE2O4 was not justified by any density-functional theory (DFT), which would provide insight into all underlying coupling constants, crystal-field parameters as well as local anisotropy axes. The studied model thus represents just a very crude and oversimplified description of physical reality. The high total angular momentum and strong crystal-field anisotropy are not sufficient for the Ising-model description.
5. Two crystallographically inequivalent positions of rare-earth ions in SrRE2O4 and BaRE2O4 imply two inequivalent local anisotropy axes, which means that the application of the magnetic field along one of them will have longitudinal as well as transverse projection with respect to the second one. This fact is not accounted for within the present model at all.
6. An infinite degeneracy of some ground states does not immediately imply their disordered character, see for instance the book by R. Liebmann, Statistical Mechanics Of Periodic Frustrated Ising Systems. For instance, the ground states with "one-dimensional disorder" often exhibit a spontaneous long-range order with nonzero critical temperature, so the "disordered" acronym is not appropriate.
7. The results presented in this work are not truly exact and they should verified by some independent numerical method (e.g. Monte Carlo simulations).
8. Some statements are too oversimplified such as:

"However, among frustrated magnets, there are many compounds with large-moment magnetic atoms. These magnets can be well described with classical Heisenberg spin models." - it is not true, even the magnetic compounds involving spin-7/2 Gd3+ ions cannot be satisfactorily described by classical (vector) spins.

"If, in addition, there are easy axes of magnetization, then Ising-type models could be applied." - it is not true, the crystal-field effects are usually deciding whether or not the Ising description is applicable.

In summary, the manuscript brings incomplete solution for a ground-state problem of the spin-1/2 Ising model on a 3D honeycomb zigzag-ladder lattice, whereas the studied model represents just an oversimplified description of rare-earth magnets SrRE2O4 and BaRE2O4. In my opinion, the present manuscript does not meet general acceptance criteria of a highly reputable journal such as Scipost Physics and I cannot recommend its publication.

---

## Round 1 · Referee Report · Anonymous (Referee 2) · 2021-8-9

Strengths

1- Detailed ground-state investigation. 2- Lots of nice figures.

Weaknesses

1- Lengthy but nevertheless incomplete. 2- Oversimplified model for the compounds mentioned in the title.

Report

An incomplete exact study is performed of the ground-state problem for an Ising model on zigzag-ladders arranged in a 3D honeycomb geometry. 5 coupling constants and 2 independent magnetic fields are considered, resulting in a correspondingly rich (again incomplete) 7-dimensional phase diagram.

I have two major concerns.

The first one is the motivation drawn from the SrRE$_2$O$_4$ and BaE$_2$O$_4$ family of compounds. These are complicated compounds resulting in the corresponding complex 7-parameter model. Given that one can get an infinite number of phases already in a relatively simple one-dimensional Ising model [see, e.g., Per Bak and R. Bruinsma, Phys. Rev. Lett. 49, 249 (1982)], it is not a surprise that a complicated 3D model gives rise to a complex phase diagram. More importantly, however, none of the SrRE$_2$O$_4$ and BaE$_2$O$_4$ compounds is probably accurately described by the model (1). Either the degrees of freedom are actually not $\pm 1$ Ising degrees of freedom, or if they are, the local axes on the "red" and "blue" sites are actually not parallel, but rather orthogonal (as stated, e.g., at the end of the second paragraph of the Introduction). Thus, the physically relevant model should include at least a certain amount of quantum fluctuations that might either smear out certain features, or stabilise ground states by a quantum order-by-disorder mechanism. By contrast, the purely classical Ising model investigated in the present work is probably not appropriate for any of the SrRE$_2$O$_4$ or BaE$_2$O$_4$ compounds.

Secondly, this manuscript is long, but still not self-contained. For example, the triangular plaquettes as building blocks are well explained, but I got the impression that the "basic rays" are not completely explained even though they are a central tool of the present investigation, nor is the "${\mathbf r}$" notation properly introduced.

There is obviously an incredible amount of work in the many nice figures of the present manuscripts. Nevertheless, the authors should not try to publish their notebooks, at least not as a journal article. There is probably a core of interesting material in the present manuscript, but it would have to be presented in a more concise and accessible manner. Furthermore, SciPost Physics Core may be a more appropriate venue even for such a rewritten manuscript.

Requested changes

1- Shorten the core manuscript such as to focus on the essential findings. 2- Discuss shortcomings of the model (1) as applied to the SrRE$_2$O$_4$ and BaE$_2$O$_4$ compounds. Removal of emphasis on these compounds (e.g., from the title) and/or focus on results that are really expected to be relevant to these compounds would also be appropriate. 3- Many details are difficult to grasp by the reader. This is particularly true for the "SUPPLEMENT" (Ref. [29]) that has a long list of tables without any explanatory text. Note also that SciPost itself does not publish Supplementary Material, but that authors should make any such material available themselves, e.g., via institutional repositories, see https://scipost.org/submissions/author_guidelines . Complementary, but non-essential information can be provided in this form. 4- Add proper explanation of basic rays. 5- The "(incomplete)" after "exact solution" at the beginning of the abstract is honest, but not very convincing. The authors might try to present this differently. 6- Fix incorrect cross-references such as to "Section H" on page 3, Figs. 9-13 instead of 9-14 (?) in the first paragraph of section 2.7 on page 14 and "Section II, Subsection F" at the beginning of the SUPPLEMENT.

---

## Round 1 · Referee Report · Anonymous (Referee 3) · 2021-8-10

Report

The paper by Dublenych and Petrenko reports on a theoretical study of an Ising model aimed to explain experimental properties of Sr(Ba)Re2O4 magnetic materials with so-called honeycomb zigzag-ladder lattice.The member compounds exhibit a variety of unusual properties including magnetization jumps, multiple phases, partial long-range order etc. To my knowledge there was little if any effort to understand these behaviors at the microscopic level. Thus the manuscript addresses a potentially interesting problem and may be relevant to SciPost. However, I find that the paper is poorly written to the extent it's hard to judge its scientific merit and relevance to the family of honeycomb zigzag-ladder materials.

Here is the list of specific remarks and comments. 1) the formulated Ising model is not justified in relation to specific SrRe2O4 or BaRe2O4 compounds. For example, Er, Dy, Tb and Ho ions may have quite different properties, Kramers vs non-Kramers, Ising vs XY. Furthermore, there are two positions for the rare-earth ions. What are the differences -- known or/and supposed, do they have different local environments, g-factors, crystal-field level structures the reader is left in dark about all that. What is the physical meaning of the effective fields h1 and h2. Are these different components of an applied magnetic field or it's the same component multiplied by the different g-factors?

2) Figure 1 showing the lattice structure is not sufficiently clear. It's just an adopted poor quality version of the Figure from Ref.[8]. Authors can show several projections along different directions and have more discussion of the choice of exchange parameters.

3) The most important comment is that the authors give, in my view, an unacceptable presentation of their theoretical approach. There are only vague and short explanations in the end of Sec.2.1, which are compensated with very long and perhaps not so necessary pictures for all configurations in the supplemental material. For details of their approach the authors refer to the previous works of one of them. This is not sufficient. Furthermore, in his early works PRE 2011, Dublenych cites the article by Kaburagi and Kanamori, JPSJ 44, 718 (1978) as the origin of the used theoretical method. The authors have to cite this work in their paper, explain it and contrast with their own approach.

To summarize, the manuscript is not worth of publishing in the present form. Still the authors have chances to improve it by making substantial revisions responding to the above remarks.

---

## Round 2 · Referee Report · Anonymous (Referee 3) · 2022-6-7

Report

The authors have significantly improved on the presentation clarity in the revised version. Taking into account an ongoing interest in the addressed experimental systems I can recommend this manuscript for publication without further delay.

---

## Round 2 · Referee Report · Anonymous (Referee 1) · 2022-6-14

Report

Reviewer report on the revised version of the manuscript scipost_202107_00007v2 entitled "An Ising model on a 3D honeycomb zigzag-ladder lattice: a solution to the ground-state problem" by Yu. I. Dublenych and O. A. Petrenko I have carefully read the revised version of the manuscript and I have to keep my previous standpoint. In my opinion the present manuscript does not meet high standards of SciPost Physics journal, because it does meet any of the four required acceptance criteria: - it does not provide a groundbreaking theoretical/experimental/computational discovery - it does not provide a breakthrough on a previously-identified and long-standing research stumbling block - it does not open a new pathway in an existing or a new research direction with clear potential for multipronged follow-up work - it does not provide a novel and synergetic link between different research areas. On the other hand, the revised manuscript provides solid paper acceptable in less prestigious journals and I would warmly recommend it for publication in SciPost Physics Core journal.

As far as my previous comments are concerned, some of the points were just simply ignored and the manuscript has not been improved in a few points anyhow.

  1. The solution for a ground-state problem of the spin-1/2 Ising model on a 3D honeycomb zigzag-ladder lattice is incomplete and hence, one cannot exclude possibility of overlooking some additional ground state(s) and overall incompleteness of the established ground-state phase diagrams.

There is no a general algorithm for solving ground-state problem for Ising-like models as there are no complete solutions even for some apparently simple problems of this kind. For instance, the model with interactions up to third neighbors on a triangular lattice [M. Kaburagi, J. Kanamori, Ground State Structure of Triangular Lattice Gas Model with up to 3rd Neighbor Interactions, J. Phys. Soc. Jpn. 44, 718 (1978)] is still not completely solved. However, even an incomplete solution is useful, especially since some regions are fully established.

I do not declare that incomplete exact solutions are not useful, but of course, they are much less valuable in comparison with complete exact solutions.

  1. The used method has been developed by one of the present authors more than 10 years ago and because of its obscurity it did not find any followers yet (at least to the best of my knowledge). From this perspective, it seems to be of marginal interest for potential readers from the methodological point of view.

One important aspect of the method used is that it represents a unique way to give absolutely complete solution of ground-state problems for Ising-like models. Even such a simple problem as ground states for the Ising model on a triangular lattice with first and second neighbor interaction in an external field was at last completely solved due to this method [see Appendix of Phys. Rev E 84, 011106 (2011)]. It is not true that the method did not find any followers. To give just one example, a PhD thesis from the Massachusetts Institute of Technology (W. Huang, “Ground state determination, ground state preserving fit for cluster expansion and their integration for robust CE construction” 2018) has 9 citations to the method and it states that: “The "basic ray" method have been successfully applied in finding devil's step (infinitely many ground states) in triangular lattice with interaction up to 3rd nearest neighbor. We mentioned that it provides a complete ground state solution to triangular lattice with interaction up to 2nd nearest neighbor and a triplet term. This method is also applied to completely resolve the ground state problem of Shastry-Sutherland lattice involving edge and diagonal interactions. Finally, it has been successfully applied to solve completely the anisotropic triangular lattice with nearest neighbor interaction.” As a side note, we find this comment less useful compared to the rest and it is actually bordering the limits of politeness. We are scientists, not religious preachers; our motivation is to find a solution to a particular problem rather than to convince as many followers as possible in the validity of the methods used. To declare the “marginal interest” on the basis of not finding the “followers” after 10 years seems to be rather short-sighted. Could it be that more time is required? Could it also be that the approach used is too laborious and therefore currently “unpopular”, but with a little help from a computer science specialist it might be transformed into a much more user-friendly routine? How does the recommendation not to publish help to improve the situation with (the harshly labelled) “obscurity”?

Well, the used method is apparently less useful for the studied model as it does not provide true and full exact solution in contrast to the example mentioned in the response to my comment, where the full exact solution is achieved. I am really sorry if the authors feel that my previous comment was at the border of politeness, it was not definitely meant in profane way. However, it is just simple fact that there are no follow-up papers by independent researchers exploiting this method (except the mentioned PhD thesis) and I have found cca. 10 000 hits in CC database for the last 10 years having the Ising model as keyword in title or abstract. Do the authors really mean that more time is needed for application of their method? The topics is really not of marginal interest. I think that it would really help if the authors would develop computer code exploiting algorithm based on the basic ray method, whereby open-source code would greatly promote application of this calculation scheme.

  1. The investigated model was developed for rare-earth magnets SrRE2O4 and BaRE2O4, however, the paper contains only a little new information about these compounds. The authors merely predict character of the ground states, but their stability (field range) is not comprehensively compared with available experimental data. The true nature of ground state is also questionable.

Please see the response to all three Referees above. The task of comprehensively comparing the ground states of a large family of compounds to the theoretical results is not realistic, given the variety of magnetic properties displayed by the family members. Section 3 of the manuscript has been revised to provide a detailed description of the family members with the most pronounced Ising-like character.

Although the revised section 3 contains a few more details on the rare-earth magnets SrRE2O4 and BaRE2O4, however, it is still more or less compilation of the previous experimental findings and a few statements concerning the nature of the ground states according to the studied models. If there is no quantitative accordance related to a stability the magnetic-field range of the relevant ground states, these statements are not on solid grounds and the nature of ground states remains unclear.

  1. The appropriateness of the suggested model for rare-earth magnets SrRE2O4 and BaRE2O4 was not justified by any density-functional theory (DFT), which would provide insight into all underlying coupling constants, crystal-field parameters as well as local anisotropy axes. The studied model thus represents just a very crude and oversimplified description of physical reality. The high total angular momentum and strong crystal-field anisotropy are not sufficient for the Ising-model description.

There are no DFT results available for the magnets discussed. This manuscript does not aim to substitute or replace them, the aim is completely different. Please see the response to all three Referees above. The Ising-like character for several members of the SrRE2O4 and BaRE2O4 families has been experimentally determined as detailed in Section 3.

The response that there is hope that at least representative of the SrRE2O4 and BaRE2O4 families is true experimental realization of the studied model appears for me more hope than reality. The modeling is more complex and the authors should consult experts in DFT methods, which would support their findings. Otherwise the statements are of speculative nature.

  1. Two crystallographically inequivalent positions of rare-earth ions in SrRE2O4 and BaRE2O4 imply two inequivalent local anisotropy axes, which means that the application of the magnetic field along one of them will have longitudinal as well as transverse projection with respect to the second one. This fact is not accounted for within the present model at all.

In the proposed model, the assumption is made that the anisotropy is sufficiently strong for both positions. This presumption allows us to consider an Ising-like model with two different fields acting on the two sites and to neglect the influence of transverse projections of the magnetic field.

This holds just if g-factor in both transverse projections are perfectly or at least nearly zero. However, this is not true in most of the rare-earth compounds where transverse projections of g-factors cannot be simply ignored.

  1. An infinite degeneracy of some ground states does not immediately imply their disordered character, see for instance the book by R. Liebmann, Statistical Mechanics Of Periodic Frustrated Ising Systems. For instance, the ground states with "one-dimensional disorder" often exhibit a spontaneous long-range order with nonzero critical temperature, so the "disordered" acronym is not appropriate.

This is a valid comment and we fully agree with the Referee, however, our analysis is restricted to only zero temperature.

This comment was slightly clarified by specifying more exactly one-dimensional or two-dimensional disorder even if I still miss the comment that the phase with one-dimensional disorder can be spontaneously ordered. From this perspective, I would not call these phases disordered at all.

  1. The results presented in this work are not truly exact and they should verified by some independent numerical method (e.g. Monte Carlo simulations).

All the results in our work are exact and can be verified analytically. Incompleteness of some diagrams is due to incompleteness of the solution. The task of numerically verifying the results (with the MC or other simulations) should be treated as a completely separate project for at least two main reasons: a) the complexity of systems considered and the number of independent parameters required to adequately model them; b) the volume of the manuscript, 30 pages, is already above optimal (see the comments from the Third Referee below), additional ~10 pages required to present/analyze the MC results will make it unreadable.

If the MC results would corroborate the reported findings, there is no need to extend the manuscript by 10 pages. All ground states are already presented in this version of the manuscript (no new figure would be needed) and one could use just a few (2-4) typical MC plots in order to verify the correctness of the established phase diagram. I am sure that 2 pages would be more than enough. Unfortunately, the authors do not show any willingness to support and validate their results.

To conclude, the present manuscript does not meet high standards of SciPost Physics journal and I suggest to reject it. I would recommend the present version of the manuscript for publication in SciPost Physics Core journal.

---

## Round 2 · Referee Report · Anonymous (Referee 2) · 2022-7-11

Report

The revised version has improved with respect to the previous one. Nevertheless, some issues still remain.

Most importantly, I do not think that the acceptance criteria for SciPost Physics (see https://scipost.org/SciPostPhys/about#criteria) are satisfied. By contrast, I think that those of SciPost Physics Core (see https://scipost.org/SciPostPhysCore/about#criteria) should be Ok.

In particular, the application to the SrRe$_2$O$_4$ and BaRE$_2$O$_4$ remains a bit far-fetched, but without these, the work is of limited relevance. I understand that the authors made an effort to motivate the Ising model better. Still, I think that one would need a transverse field that is small as compared to the longitudinal field, a condition that in my opinion is impossible to satisfy for almost orthogonal axes since then at least one transverse field will be at least almost as big as its longitudinal component.

I have some further concrete comments that I list as "Requested changes".

Requested changes

1- I appreciate the change of title and abstract, as requested. However, now a reader will probably wonder about the motivation for studying a seven-parameter model, as announced in the first sentence of the abstract. It would probably be better to motivate this from the experimental situation even if it is a strong idealization thereof. 2- Line 3 of section 2.2: I think the authors mean "vertices" (not "vortexes"). 3- The "${\bf r}$" notation (first column of Table 1) seems to be still unexplained. 4- With respect to a comment of another Referee: it might indeed be better to replace the term "disorder" by "degeneracy". 5- Line 4 of section 2.6: correct spelling of "incomplete". 6- There is very little discussion of Fig. 14 in the text, in particular none of its physical content. 7- Format Eq. (7) properly with the LaTeX “cases” environment. 8- On applicability of the Ising model: I think that more comments are needed than just stating that there are two inequivalent sites. This concerns in particular the second paragraph of section 3, but also the first paragraph of section 2.1. 9- Align Eq. (16) properly to ensure readability. 10- Upper-case "Ising" in the title of Ref. [10]. 11- Fix chemical formulas in the title of Ref. [36].

---

## Round 2 · Author Response

Dear Editor,

Thank you for considering our manuscript. Following the referees’ comments and recommendation, we have substantially revised the manuscript. We believe that the measures taken have significantly improved the manuscript’s presentation. We are also convinced that the manuscript presents important theoretical results applicable to many experimental systems and therefore would kindly ask you to consider it again for publication in SciPost Physics.

Common response to all the three Referees.

We are genuinely grateful to all three Referees for their careful reading of the manuscript, for their critical assessment of it, as well as for their very helpful suggestions. We provide a detailed point-by-point reply to all the comments made; however, we believe it is helpful to start our reply to the referees’ criticism with a short general statement. We are the first to admit that the manuscript under consideration is not very “conventional”, as it contains a rather large number of multi-panel figures and bulky tables with plenty of unusual looking symbols introduced. We tried very hard to make the manuscript as concise as possible, but this (fully justifiable) requirement from the referees contradicts another essential demand – to provide a complete description of the proposed solution and the methods used to obtain it. We hope that the revised version of the manuscript gets the delicate balance right, but if the referees are still not entirely happy with it, we are open to the suggestions on further improvements of the structure. An important remark here is about the nature of the manuscript, which is largely (if not entirely) theoretical, but still it is motivated by the recent experimental findings for the family of the SrRE2O4 and BaRE2O4 compounds. All referees seem to agree that the family “exhibits a variety of unusual properties” and is “definitely worthwhile to study.” There is a strong experimental evidence, please see the discussion below, that at least in SrEr2O4, SrHo2O4, SrDy2O4, BaDy2O4 (and perhaps other family members) the magnetic interactions are highly anisotropic in nature. It is indeed an oversimplification to call these systems purely Ising, as a full description of the magnetic properties requires the introduction of the exchange interactions, dipole-dipole interactions and the effects of rather complex (and variable) crystal fields interactions. Given the geometry of the lattice and the presence of the two inequivalent sites, it is not surprising to find that, as absolutely correctly pointed out by one of the referees, “there was little if any effort to understand these behaviors at the microscopic level”. The main aim of our manuscript could be phrased as follows. Let us presume that at least some of the SrRE2O4 family members can be described as Ising magnets. What is then the solution for the Ising model on a 3D honeycomb zigzag-ladder lattice? In order to answer this question, we use, expand and adopt a particular method, previously employed for different geometries. It turns out to be a non-trivial, yet rather fascinating exercise, which returned (perhaps rather surprisingly) several results directly applicable to the SrRE2O4 compounds.

Response to the First Referee.

  1. The solution for a ground-state problem of the spin-1/2 Ising model on a 3D honeycomb zigzag-ladder lattice is incomplete and hence, one cannot exclude possibility of overlooking some additional ground state(s) and overall incompleteness of the established ground-state phase diagrams.

There is no a general algorithm for solving ground-state problem for Ising-like models as there are no complete solutions even for some apparently simple problems of this kind. For instance, the model with interactions up to third neighbors on a triangular lattice [M. Kaburagi, J. Kanamori, Ground State Structure of Triangular Lattice Gas Model with up to 3rd Neighbor Interactions, J. Phys. Soc. Jpn. 44, 718 (1978)] is still not completely solved. However, even an incomplete solution is useful, especially since some regions are fully established.

  1. The used method has been developed by one of the present authors more than 10 years ago and because of its obscurity it did not find any followers yet (at least to the best of my knowledge). From this perspective, it seems to be of marginal interest for potential readers from the methodological point of view.

One important aspect of the method used is that it represents a unique way to give absolutely complete solution of ground-state problems for Ising-like models. Even such a simple problem as ground states for the Ising model on a triangular lattice with first and second neighbor interaction in an external field was at last completely solved due to this method [see Appendix of Phys. Rev E 84, 011106 (2011)]. It is not true that the method did not find any followers. To give just one example, a PhD thesis from the Massachusetts Institute of Technology (W. Huang, “Ground state determination, ground state preserving fit for cluster expansion and their integration for robust CE construction” 2018) has 9 citations to the method and it states that: “The "basic ray" method have been successfully applied in finding devil's step (infinitely many ground states) in triangular lattice with interaction up to 3rd nearest neighbor. We mentioned that it provides a complete ground state solution to triangular lattice with interaction up to 2nd nearest neighbor and a triplet term. This method is also applied to completely resolve the ground state problem of Shastry-Sutherland lattice involving edge and diagonal interactions. Finally, it has been successfully applied to solve completely the anisotropic triangular lattice with nearest neighbor interaction.” As a side note, we find this comment less useful compared to the rest and it is actually bordering the limits of politeness. We are scientists, not religious preachers; our motivation is to find a solution to a particular problem rather than to convince as many followers as possible in the validity of the methods used. To declare the “marginal interest” on the basis of not finding the “followers” after 10 years seems to be rather short-sighted. Could it be that more time is required? Could it also be that the approach used is too laborious and therefore currently “unpopular”, but with a little help from a computer science specialist it might be transformed into a much more user-friendly routine? How does the recommendation not to publish help to improve the situation with (the harshly labelled) “obscurity”?

  1. The investigated model was developed for rare-earth magnets SrRE2O4 and BaRE2O4, however, the paper contains only a little new information about these compounds. The authors merely predict character of the ground states, but their stability (field range) is not comprehensively compared with available experimental data. The true nature of ground state is also questionable.

Please see the response to all three Referees above. The task of comprehensively comparing the ground states of a large family of compounds to the theoretical results is not realistic, given the variety of magnetic properties displayed by the family members. Section 3 of the manuscript has been revised to provide a detailed description of the family members with the most pronounced Ising-like character.

  1. The appropriateness of the suggested model for rare-earth magnets SrRE2O4 and BaRE2O4 was not justified by any density-functional theory (DFT), which would provide insight into all underlying coupling constants, crystal-field parameters as well as local anisotropy axes. The studied model thus represents just a very crude and oversimplified description of physical reality. The high total angular momentum and strong crystal-field anisotropy are not sufficient for the Ising-model description.

There are no DFT results available for the magnets discussed. This manuscript does not aim to substitute or replace them, the aim is completely different. Please see the response to all three Referees above. The Ising-like character for several members of the SrRE2O4 and BaRE2O4 families has been experimentally determined as detailed in Section 3.

  1. Two crystallographically inequivalent positions of rare-earth ions in SrRE2O4 and BaRE2O4 imply two inequivalent local anisotropy axes, which means that the application of the magnetic field along one of them will have longitudinal as well as transverse projection with respect to the second one. This fact is not accounted for within the present model at all.

In the proposed model, the assumption is made that the anisotropy is sufficiently strong for both positions. This presumption allows us to consider an Ising-like model with two different fields acting on the two sites and to neglect the influence of transverse projections of the magnetic field.

  1. An infinite degeneracy of some ground states does not immediately imply their disordered character, see for instance the book by R. Liebmann, Statistical Mechanics Of Periodic Frustrated Ising Systems. For instance, the ground states with "one-dimensional disorder" often exhibit a spontaneous long-range order with nonzero critical temperature, so the "disordered" acronym is not appropriate.

This is a valid comment and we fully agree with the Referee, however, our analysis is restricted to only zero temperature.

  1. The results presented in this work are not truly exact and they should verified by some independent numerical method (e.g. Monte Carlo simulations).

All the results in our work are exact and can be verified analytically. Incompleteness of some diagrams is due to incompleteness of the solution. The task of numerically verifying the results (with the MC or other simulations) should be treated as a completely separate project for at least two main reasons: a) the complexity of systems considered and the number of independent parameters required to adequately model them; b) the volume of the manuscript, 30 pages, is already above optimal (see the comments from the Third Referee below), additional ~10 pages required to present/analyze the MC results will make it unreadable.

  1. Some statements are too oversimplified such as: "However, among frustrated magnets, there are many compounds with large-moment magnetic atoms. These magnets can be well described with classical Heisenberg spin models." - it is not true, even the magnetic compounds involving spin-7/2 Gd3+ ions cannot be satisfactorily described by classical (vector) spins. Our reading of the above comment is that in some cases even a large spin (such as 7/2 in Gd3+) does not guaranty the validity of a classical approach. This is absolutely true, however, there are many systems with a large spin for which classical description is adequate. The revised version of the manuscript reads “However, among frustrated magnets, there are many compounds with large-moment magnetic atoms. Often, these magnets can be well described with classical, either Heisenberg or Ising spin models, depending on the presence of strong crystal-field effects.”

"If, in addition, there are easy axes of magnetization, then Ising-type models could be applied." - it is not true, the crystal-field effects are usually deciding whether or not the Ising description is applicable. In the revised version of the manuscript, we write “strong crystal-field effects” instead of “easy axes of magnetization.”

Response to the Second Referee. 1) the formulated Ising model is not justified in relation to specific SrRe2O4 or BaRe2O4 compounds. For example, Er, Dy, Tb and Ho ions may have quite different properties, Kramers vs non-Kramers, Ising vs XY. Furthermore, there are two positions for the rare-earth ions. What are the differences -- known or/and supposed, do they have different local environments, g-factors, crystal-field level structures the reader is left in dark about all that. What is the physical meaning of the effective fields h1 and h2. Are these different components of an applied magnetic field or it's the same component multiplied by the different g-factors?

Most of the questions asked are addressed in Section 3 of the manuscript. Since there are two nonequivalent positions for the rare-earth ions, h1 and h2 are different components (projections) of an applied magnetic field multiplied by different g-factors.

2) Figure 1 showing the lattice structure is not sufficiently clear. It's just an adopted poor quality version of the Figure from Ref.[8]. Authors can show several projections along different directions and have more discussion of the choice of exchange parameters.

We have replaced Fig. 1 in the revised version of the manuscript. Combined with Fig. 2, they give a full description of the crystal structure. As for the choice of the exchange parameters, we consider all the possible values of these and restrict ourself to the absolute minimum number.

3) The most important comment is that the authors give, in my view, an unacceptable presentation of their theoretical approach. There are only vague and short explanations in the end of Sec. 2.1, which are compensated with very long and perhaps not so necessary pictures for all configurations in the supplemental material. For details of their approach the authors refer to the previous works of one of them. This is not sufficient. Furthermore, in his early works PRE 2011, Dublenych cites the article by Kaburagi and Kanamori, JPSJ 44, 718 (1978) as the origin of the used theoretical method. The authors have to cite this work in their paper, explain it and contrast with their own approach.

We have addressed the presentation of the theoretical methods used in the revised version of the manuscript. To find ground states of Ising-like lattice models, Kaburagi and Kanamori developed a method of geometrical inequalities. We elaborated a new method that we called “method of basic rays and basic sets of cluster configurations.” This is a cluster method that can be considered as a development and modification of a cluster method proposed by D. Lyons and T. Kaplan (Ref. [19]). In the revised version of manuscript we cite reference 21: “… in particular the method of geometric inequalities by J. Kanamori and M. Kaburagi [20,21], no general algorithm exists to obtain the ground-state phase diagram.”

Response to the Third Referee.

  1. Shorten the core manuscript such as to focus on the essential findings.

There is no Supplementary Material and section “Completeness of sets of basic rays for each phase” in the revised version of the manuscript; there is only much shorter Appendix B instead. Unfortunately, we cannot further shorten the core manuscript for two reasons: 1) we give an exact solution of the ground-state problem in such a way that readers could verify it; 2) the Hamiltonian contains seven parameters and therefore the solution is cumbersome. A clear structure of the revised manuscript will be helpful for those readers who are primarily interested in an application of the theory to experimental problems, as after reading Section 1 Introduction and perhaps subsection 2.1 Model and method, they could skip most of technical detail and proceed to Section 3 Application to SrRE2O4 and BaRE2O4 compounds.

  1. Discuss shortcomings of the model (1) as applied to the SrRE2O4 and BaE2O4 compounds. Removal of emphasis on these compounds (e.g., from the title) and/or focus on results that are really expected to be relevant to these compounds would also be appropriate.

We followed the advice, modified the title and expanded the manuscript in several places to emphasize the correspondence between the theoretical results and the experimental observations.

  1. Many details are difficult to grasp by the reader. This is particularly true for the "SUPPLEMENT" (Ref. [29]) that has a long list of tables without any explanatory text. Note also that SciPost itself does not publish Supplementary Material, but that authors should make any such material available themselves, e.g., via institutional repositories, see https://scipost.org/submissions/author guidelines . Complementary, but non-essential information can be provided in this form.

We have restricted the manuscript removing the Supplementary Material and moving some nonessential information into the Appendixes.

  1. Add proper explanation of basic rays.

We have extended the explanation of the method used and of the basic rays, in particular.

  1. The "(incomplete)" after "exact solution" at the beginning of the abstract is honest, but not very convincing. The authors might try to present this differently.

We have significantly modified the abstract following the advice.

  1. Fix incorrect cross-references such as to "Section H" on page 3, Figs. 9-13 instead of 9-14 (?) in the first paragraph of section 2.7 on page 14 and "Section II, Subsection F" at the beginning of the SUPPLEMENT.

We fixed the incorrect cross-references.

Sincerely yours, Yuriy Dublenych.

---

## Round 2 · List of Changes

We have substantially revised the manuscript.
Many changes have been made.

---

## Round 3 · Referee Report · Anonymous (Referee 2) · 2022-8-3

Weaknesses

The application to the Sr$RE_2$O$_4$ and Ba$RE_2$O$_4$ compounds may remain controversial, but this is well document in previous reports.

Report

I quickly went over the revised manuscript and would like to thank the authors for yet another round of revisions. It is a pleasure to now recommend publication in SciPost Physics Core. Nevertheless, a few further details remain that the authors might still wish to take into consideration, see "Requested changes".

Requested changes

The following items of my previous report have not or only partially been addressed: 1- The title is good. However, the abstract no longer refers to the compounds. Without these, I see no real motivation for considering "a complex, seven-parameter ground-state problem for an Ising model on a 3D honeycomb zigzag-ladder lattice, containing two types of magnetic sites, ... in the presence of an external field". Furthermore, for such a complex seven-parameter model, "the emergence of a large variety of magnetic phases" may not come as a surprise. I therefore invite the authors once again to reconsider the abstract. 3- I still have the feeling that a remark like that the "${\bf r}$" are labels for basic rays is missing. 7- Eq. (7) has improved, but I still think that this would be a case for a LaTeX "cases" environment.

---

## Round 3 · Author Response

Dear Editor,

We made almost all minor revisions to the manuscript requested by the Third Referee.

Sincerely yours,
Yuriy Dublenych.

---

## Round 3 · List of Changes

We did not change the title since the paper provides sufficient motivation in the abstract as well as in the main text.
We corrected the errors pointed out by the Third Referee and added a few sentences.

Section 2.3
Basic vector $\mathbf{r}_{1}^{\star}$, for instance, and the corresponding basic set of cluster configurations can be obtained from vector $\mathbf{r}_{1}$ and its basic set of cluster configurations by using the ${}^\star$ transformation.

Section 2.7
When both the interaction parameter values and the external fields, $h_1$ and $h_2$, are fixed we have a single point on the phase diagram. By continuously varying the value (and possibly the direction) of an external field, instead of a point, we generate a line of transitions on the phase diagram. For instance, increasing magnetic field along the $a$ axis in SrEr$_2$O$_4$ that corresponds to a passage along the $h_1$ axis in Fig.~\ref{fig14} (left middle panel) we have the following sequence of the phases: 13, 23, 15, $\tilde 6$ and 4 (see also Fig.~\ref{fig15}).

---

## Editorial Decision

published